# Quantum Algorithms for Triangle Cut Sparsification

Shan Jiang [1]  Pan Peng [1]

## Abstract

Triangles capture higher-order structures in graphs and are fundamental to applications such as clustering and network analysis. To enable efficient use of such structures at scale, we study the problem of *triangle cut sparsification*, which aims to reduce the graph size while approximately preserving triangle counts across every cut. We investigate *quantum algorithms* for this problem, using triangle listing as our main technical ingredient. In particular, we present a quantum algorithm for triangle listing that, for a graph with $n$ vertices, $m$ edges, and $t$ triangles, runs in time $T_{\text{q-list}} = \widetilde{O}\big(\min(n^{5/4}t^{7/12} + n^{7/6}t^{7/9}, m + m^{3/4}t^{1/2}, n^{3/2}t^{1/2})\big)$, improving upon the best known classical bounds over a broad range of parameters. Our algorithm is based on a heavy-light vertex partition and an extension of triangle detection via quantum walks and Grover search. Leveraging this result, we design a quantum algorithm for constructing $\varepsilon$-triangle cut sparsifiers of size $\tilde{O}(n/\varepsilon^2)$ in time $\widetilde{O}(T_{\text{q-list}} + \sqrt{mn}/\varepsilon)$. Finally, we demonstrate applications to clustering algorithms based on triangle-related measures and prove a lower bound of $\Omega(n/\varepsilon^2)$ on the size of any $\varepsilon$-triangle cut sparsifiers.

## 1. Introduction

Graph sparsification reduces the computational and memory costs of large graphs while approximately preserving important structural properties. Classical examples include graph spanners (Chew, 1986), cut sparsifiers (Benczúr & Karger, 1996), and spectral sparsifiers (Spielman & Srivastava, 2008; Batson et al., 2009). These notions have been extended naturally to hypergraphs, yielding corresponding cut and spectral sparsifiers (Chen et al., 2020a; Soma & Yoshida, 2019).

Graph sparsification has wide applications in algorithms and machine learning, including network flows (Chen et al., 2025a), computer vision (Simonovsky & Komodakis, 2017), and streaming learning (Laenen & Sun, 2020).

Recently, Kapralov et al. (2022) introduced *motif cut sparsifiers*, which sparsify a graph while approximately preserving the number of motifs (i.e., fixed subgraphs) crossing every cut. This notion is motivated by the importance of higher-order structures in graph machine learning, with applications in social network analysis (Newman, 2003; Guo et al., 2023; Hu et al., 2023), recommendation systems (Tsourakakis et al., 2011; Jiang et al., 2022), and graph clustering (Austin R. Benson & Leskovec, 2016; Li et al., 2017; Ma et al., 2019; Shi et al., 2020). Their algorithm achieves linear-size sparsifiers but relies on explicit motif enumeration, incurring high computational cost. The framework generalizes classical cut and hypergraph sparsifiers and has been extended to privacy-preserving settings for triangle motifs (Peng & Xu, 2025). In this work, we focus on *triangle cut sparsifiers* and aim to improve the efficiency of their construction.

Quantum computing offers a promising avenue for accelerating machine learning tasks and graph algorithms. In the context of sparsification, Apers & De Wolf (2022) gave the first quantum algorithm for constructing linear-size spectral sparsifiers, achieving a running time of[1] $\widetilde{O}(\sqrt{mn}/\varepsilon)$, and raised open questions about extending their speedup techniques to broader sparsification tasks. Liu et al. (2025) later developed a quantum algorithm for hypergraph sparsification, obtaining an $\varepsilon$-spectral sparsifier in time $\widetilde{O}(r\sqrt{mn}/\varepsilon)$ for a hypergraph of rank $r$. On the lower-bound side, a quantum query lower bound of $\widetilde{\Omega}(\sqrt{mn}/\varepsilon)$ is known for the edge case (Apers & De Wolf, 2022), and an analogous lower bound of $\Omega(r\sqrt{mn}/\varepsilon)$ has been conjectured for hypergraph sparsification (Liu et al., 2025). Together, these results demonstrate that quantum techniques can substantially reduce the cost of sparsifier construction, yet triangle cut sparsification—a natural higher-order analogue—has remained unexplored from a quantum perspective.

Motivated by this gap, we develop quantum algorithms for triangle cut sparsification that leverage efficient quantum triangle listing, enabling faster sparsification procedures as well as applications to triangle-based clustering.

---

[1]School of Computer Science and Technology, University of Science and Technology of China, Hefei, China. Correspondence to: Pan Peng <ppeng@ustc.edu.cn>.

*Proceedings of the 43rd International Conference on Machine Learning*, Seoul, South Korea. PMLR 306, 2026. Copyright 2026 by the author(s).

---

[1]We use $\widetilde{O}(\cdot)$ to hide polylogarithmic factors.

## 1.1. Our Contributions

We present a quantum algorithm for constructing *triangle cut sparsifiers*. In a weighted graph $G = (V, E, w)$, the weight of a triangle instance $T = (V_T, E_T)$ is defined as $w(T) = \prod_{e \in E_T} w(e)$. For a subset $S \subseteq V$, let $(S, V \setminus S)$ denote the corresponding cut. A triangle instance $T$ is said to *cross* this cut if it has vertices on both sides, i.e., $V_T \cap S \neq \emptyset$ and $V_T \cap (V \setminus S) \neq \emptyset$.

**Definition 1.1** (Triangle Cut Sparsifier, (Kapralov et al., 2022)). Let $G = (V, E, w)$ be a weighted graph. For any $\varepsilon > 0$, a reweighted subgraph $G' = (V, E', w')$ is an $\varepsilon$-triangle cut sparsifier of $G$ if, for every cut $(S, V \setminus S)$, $(1 - \varepsilon)\operatorname{cut}_G(S, V \setminus S) \leq \operatorname{cut}_{G'}(S, V \setminus S) \leq (1 + \varepsilon)\operatorname{cut}_G(S, V \setminus S)$, where $\operatorname{cut}_G(S, V \setminus S) = \sum_{T \text{ is a triangle, } T \text{ crosses } (S, V \setminus S)} w(T)$.

We emphasize that triangle cut sparsifiers preserve the *weighted* sum of crossing triangles, not the number of triangles. For instance, let $K_n$ be the complete graph with unit edge weights. For any balanced cut $(S, \bar{S})$, $\operatorname{cut}_{K_n}(S, \bar{S}) = \Theta(n^3)$. Now consider a random $d$-regular sparsifier $H$ with $d = \Theta(1/\varepsilon^2)$ where every edge is of weight $w'(e) = \Theta(n/d)$. $H$ has $O(n/\varepsilon^2)$ edges and thus at most $O(n^{3/2}/\varepsilon^3)$ triangles. Yet for a balanced cut, the expected number of crossing triangles in $H$ is $\Theta(d^3)$, and each triangle weight is $\Theta\big((n/d)^3\big)$. Consequently, $\operatorname{cut}_H(S, \bar{S}) = \Theta\big(d^3 \cdot (n/d)^3\big) = \Theta(n^3)$, matching the original cut value. Thus, a sparse reweighted graph can effectively preserve weighted cut values even though its triangle count is far smaller.

Our algorithm has the following performance guarantee.

**Theorem 1.2.** *Given query access to an undirected weighted graph $G$ with $n$ vertices and $m$ edges, there exists a quantum algorithm that, with high probability [2], constructs an $\varepsilon$-triangle cut sparsifier of $G$ containing $\widetilde{O}(n/\varepsilon^2)$ edges. The running time of the algorithm is $T_{\text{q-list}} + \widetilde{O}(\sqrt{mn}/\varepsilon)$, where $T_{\text{q-list}}$ denotes the running time of quantum triangle listing as stated in Theorem 1.3.*

For comparison, the classical algorithm of Kapralov et al. (2022) runs in time $\widetilde{O}(\min\{T_{\text{c-list}}, n^\omega\} + m)$, where $T_{\text{c-list}}$ is the classical complexity of triangle listing (see the detailed expression below) and $\omega < 2.372$ is the matrix multiplication exponent (Alman et al., 2025). The algorithm consists of a triangle-listing phase followed by a post edge-sampling phase. Our quantum algorithm accelerates the triangle-listing phase for a broad class of graphs—including sparse graphs and those with moderate triangle counts (see the discussion below), and further reduces the post-sampling cost from $\widetilde{O}(m)$ to $\widetilde{O}(\sqrt{mn}/\varepsilon)$, yielding a polynomial im-

---

[2]Throughout the paper, we use "with high probability" to mean with probability at least $1 - \operatorname{poly}(1/n)$.

provement in this stage.

We further prove an $\Omega(n/\varepsilon^2)$ lower bound on the size of any $\varepsilon$-triangle cut sparsifiers (see Theorem 5.2), establishing near-tight sparsity and providing the first matching lower bound specialized to triangle motifs.

Now we formally state the performance guarantee of our quantum algorithm for *triangle listing*.

**Theorem 1.3.** *Given query access to an undirected graph $G$ with $n$ vertices, $m$ edges and $t$ triangles, there exists a quantum algorithm, Q-TRIANGLELISTING that, with high probability, lists all triangles in $G$ with runtime $T_{\text{q-list}} = \widetilde{O}\left(\min(n^{5/4}t^{7/12} + n^{7/6}t^{7/9}, m + m^{3/4}t^{1/2}, n^{3/2}t^{1/2})\right)$.*

We assume $t \geq 1$ throughout the paper. This result provides the first provable quantum speedup for the triangle listing problem across a broad range of graphs. The speedup stems from non-trivial quantum primitives and carefully adapting from triangle detection to listing, rather than acting as a black-box wrapper; consequently, removing these quantum components would eliminate the polynomial gains. The three terms in our runtime $T_{\text{q-list}}$ correspond to quantum walk, heavy-light vertex partition and Grover search, respectively. We compare our bound with the fastest known classical algorithms (Björklund et al., 2014), whose running time is $T_{\text{c-list}} = \widetilde{O}\left(n^\omega + n^{\frac{3(\omega-1)}{5-\omega}}t^{\frac{2(3-\omega)}{5-\omega}}\right)$ or $\widetilde{O}\left(m^{\frac{2\omega}{\omega+1}} + m^{\frac{3(\omega-1)}{\omega+1}}t^{\frac{3-\omega}{\omega+1}}\right)$ for sparse graphs. In comparison, when the number of triangles satisfies $t \leq n^{15/14}$, our running time is at most $\widetilde{O}(n^2)$, matching or improving upon the conjectured classical lower bound $\widetilde{O}(n^2 + nt^{2/3})$ obtained by setting $\omega = 2$. For sparse graphs, the term $m + m^{3/4}t^{1/2}$ based on heavy–light vertex partition dominates our running time. Assuming $\omega = 2$, the best known classical bound simplifies to $\widetilde{O}(m^{4/3} + mt^{1/3})$, which is widely believed to be conditionally optimal (Patrascu, 2010). In contrast, our bound is always no larger whenever $t \leq m^{3/2}$ (which holds for all graphs, see Fact C.6), demonstrating a quantum advantage. A discussion of the query complexity of our algorithms is provided in Appendix D.

To the best of our knowledge, no quantum lower bounds are currently known for triangle listing. Nevertheless, our algorithm encounters fundamental limitations. In particular, the $\widetilde{O}(n^{3/2}t^{1/2})$ term approaches Grover-optimality (the method cannot be improved beyond logarithmic factors). Moreover, in the two extreme regimes, our algorithm matches known optimal bounds. When $t = 1$, the running time reduces to $\widetilde{O}(n^{5/4})$, matching the best known quantum bound for triangle detection (Le Gall, 2014); when $t = \Theta(n^3)$, the running time becomes $\Theta(n^3)$, which is unavoidable whether classically or quantumly due to the inherent output size.

Finally, we show that our quantum triangle listing primitive

can be directly incorporated into several triangle-based clustering algorithms, including spectral and PageRank-based methods (Austin R. Benson & Leskovec, 2016; Tsourakakis et al., 2017; Yin et al., 2017), thereby yielding corresponding quantum speedups.

## 1.2. Technical Overview

Our quantum algorithm for triangle cut sparsification builds on the strength-based motif sparsification framework of Kapralov et al. (2022), which treats each triangle as a hyperedge and iteratively sparsifies the graph by retaining edges of high estimated importance while sampling the remaining edges. Although this framework yields a sparsifier with small size (i.e., $\widetilde{O}(n)$ edges), its classical implementation is bottlenecked by triangle enumeration and repeated $\widetilde{O}(m)$-time post-sampling, which together dominate the runtime. Our contribution is to provide quantum speedups for both components: triangle listing and post-sampling.

**Quantum Triangle Listing.** We use three approaches for triangle listing. The first approach is based on heavy–light vertex partition, which distinguishes triangles by whether they involve a low-degree vertex. Vertices are classified as light or heavy by a degree threshold $\Delta$, ensuring that light vertices have bounded neighborhoods while the number of heavy vertices remains small. Triangles incident to light vertices are listed by applying repeated Grover search over pairs of neighbors, and each light vertex is removed after processing to avoid duplication. The remaining triangles, which are induced entirely by heavy vertices, are listed via a direct Grover search over triples of heavy vertices. By balancing the costs of two phases and setting $\Delta = \sqrt{m}$, this approach lists all triangles with high probability in $\widetilde{O}(m + m^{3/4}t^{1/2})$ time.

The second approach extends Le Gall's quantum triangle detection algorithm (Le Gall, 2014), which decides the existence of a triangle using $\widetilde{O}(n^{5/4})$ queries. The algorithm samples a vertex set $S$ of size $\sqrt{n}$ and efficiently lists all triangles incident to $S$ via Grover search. The remaining task is to find triangles entirely outside $S$. Since vertex-pairs sharing a common neighbor in randomly sampled $S$ are excluded, the search space for subset $X$ is reduced from $O(|X|^2)$ to $O(|X|^2/\sqrt{n})$ with high probability. A quantum walk is then performed on the Johnson graph $J(V, \lceil n^{3/4} \rceil)$ to search for a subset containing an edge of a triangle. To check whether a subset $A$ is marked, apply a variable-cost quantum search to identify a common neighbor and an inner quantum walk on $J(A, \lceil \sqrt{n} \rceil)$. This framework naturally extends triangle detection to listing (i.e., with more triangles), but requires careful handling to avoid rediscovering the same triangle multiple times.

We address this by first listing all triangles incident to $S$ as well as all triangles sharing edges with them, then repeatedly

running a quantum walk whose marked-set fraction scales with the number of remaining triangles. This yields a total cost of $\widetilde{O}(n^{5/4}t^{7/12})$ for graphs with fewer triangles. To avoid duplicates, we introduce an auxiliary procedure that outputs all triangles sharing an edge with the newly found triangle and then removes those three edges from the oracle. This ensures progress and contributes an additional $O(\sqrt{n}t)$ term.

The third approach uses Grover search directly, achieving a runtime of $\widetilde{O}(n^{3/2}t^{1/2})$, which is preferable for dense graphs and when triangles are abundant.

**Quantum Post-Sampling.** Assuming all triangles have been listed, we next accelerate the post-sampling stage. Classically, each iteration identifies *critical edges* (with large estimated importance) and samples the remaining edges. We accelerate the first step using our quantum triangle listing algorithm, and the second step by encoding sampling decisions implicitly, recovering surviving edges only at the end via quantum search (Apers & De Wolf, 2022). Overall, the algorithm constructs an $\varepsilon$-triangle cut sparsifier with high probability in time $T_{\text{q-list}} + \widetilde{O}\left(\frac{\sqrt{mn}}{\varepsilon}\right)$, demonstrating that strength-based motif sparsification admits substantial quantum speedups.

## 1.3. Other Related Work

Triangle detection and listing are classical problems in graph algorithms and fine-grained complexity (see e.g., Itai & Rodeh (1977); Alon et al. (1997); Björklund et al. (2014); Williams & Xu (2020); Bringmann & Gorbachev (2025)).

Quantum computing has enabled substantial speedups for triangle detection. Grover's method first reduced the query complexity to $O(n^{3/2})$ (Buhrman et al., 2001), followed by a sequence of improvements using various techniques (Magniez et al., 2007b; Belovs, 2012; Lee et al., 2013; Jeffery et al., 2013). The current best bounds achieve $\widetilde{O}(n^{5/4})$ queries for unweighted graphs (Le Gall, 2014), with further improvements for sparse graphs (Le Gall & Nakajima, 2017) and logarithmic factors removed in (Carette et al., 2019). Despite this progress, the *triangle listing* problem remains largely unexplored in the quantum setting.

A more detailed discussion of classical and quantum algorithms for triangle-related problems, including other quantum advances, is provided in Appendix B.

## 2. Preliminaries

Let $G = (V, E, w)$ be an undirected graph, where $V$ is the vertex set $[n] = \{1, 2, \ldots, n\}$, $E$ is the edge set of cardinality $m$, and $w \colon E \to \mathbb{R}_{\geq 0}$ represents edge weights. For $v \in V$, we denote by $N(v)$ its neighbor set and $\deg(v)$ its degree. A *triangle* is a 3-node complete graph $K_3$. A

subgraph of $G$ that is isomorphic to a triangle is called a *triangle instance* $T$ in $G$ (or simply a *triangle* in $G$). We denote by $\mathcal{T}(G)$ the set of all triangle instances in $G$, with the triangle count in $G$ defined as $t := |\mathcal{T}(G)|$.

**Quantum Computing Background.** In quantum computing, the state of a system is represented by a unit vector $|v\rangle$ in a Hilbert space $\mathcal{H}$, which is a complex inner product vector space. For a $d$-dimensional Hilbert space $\mathbb{C}^d$, the standard computational basis consists of orthonormal vectors $\{|i\rangle\}_{i=0}^{d-1}$, where $|i\rangle = (0, \ldots, 0, 1, 0, \ldots, 0)^\top$ denotes the $i$-th basis vector. A single qubit is represented as $\alpha |0\rangle + \beta |1\rangle$ with complex amplitudes $\alpha, \beta$ satisfying $|\alpha|^2 + |\beta|^2 = 1$. Multi-qubit states are represented via tensor products: for $|a\rangle, |b\rangle \in \mathbb{C}^d$, their tensor product is their Kronecker product $|a\rangle \otimes |b\rangle \equiv |a\rangle |b\rangle = (a_0 b_0, a_0 b_1, \ldots, a_1 b_0, a_1 b_1, \ldots, a_{d-1} b_{d-1})^\top \in \mathbb{C}^{d^2}$.

The evolution of a closed quantum system is described by a unitary operator $U$ satisfying $U^\dagger U = I$, acting as $|\psi\rangle \mapsto U |\psi\rangle$, where $U^\dagger$ is the Hermitian conjugate of $U$, and $I$ is the identity matrix. To extract classical information from state $|\psi\rangle$, measurement in the computational basis yields outcome $i$ with probability $|\langle \psi | i \rangle|^2$, collapsing the state to $|i\rangle$, where $\langle \psi | i \rangle$ is the inner product of $|\psi\rangle$ and $|i\rangle$.

**Quantum Computational Model.** We assume the input graph $G = (V, E, w)$ is stored in a quantum-accessible classical memory (QRAM) that supports the following queries coherently:

- **degree query:** given $v \in V$, return its degree $\deg(v)$;
- **neighbor query:** given $v \in V$ and an index $i \in [\deg(v)]$, return the $i$-th neighbor of $v$;
- **vertex-pair query:** given a pair of vertices $\{u, v\}$, return the weight $w(u, v)$ if $\{u, v\} \in E$, and 0 otherwise.

This is equivalent to having general graph model in QRAM, as standard in quantum graph algorithms (e.g., Apers & De Wolf (2022); Chen et al. (2025b)).

Quantum algorithms are described in the quantum circuit model, operating on $|0\rangle^{\otimes n}$, applying a sequence of unitary gates and oracle queries, and ending with measurement. The time complexity of a quantum algorithm is measured by the total number of elementary gates, oracle queries, and QRAM operations. We also include involved classical steps and the cost of writing outputs (e.g., storing the list of triangles). In particular, since triangle listing outputs $t$ items, any algorithm for this task must spend $\Omega(t)$ time; our upper bounds always dominate this bound, so it does not appear as an extra term.

Although QRAM is a standard assumption, it is a stronger model than the plain circuit model where we usually consider query complexity that only counts the number of oracle queries. We remark that our guarantees are asymptotic.

While we do not provide concrete hardware crossover points, the polynomial improvements we achieve suggest that, once large-scale fault-tolerant quantum computers become available, the advantages would be relevant for massive graphs.

**Quantum Tools.** Below are some used tools.

**Lemma 2.1** (Grover search, (Grover, 1996)). *Let $g\colon [N] \to \{0, 1\}$ be a Boolean predicate and $M = \{i \in [N] \mid g(i) = 1\}$. There exists a quantum algorithm that, with probability at least $2/3$,*

- *(standard version) finds one element of $M$ (if $M \neq \emptyset$) in $\widetilde{O}(\sqrt{N/|M|})$ time;*
- *(repeated version) finds all elements of $M$ in $\widetilde{O}(\sqrt{N|M|})$ time.*

**Quantum Walk on Johnson Graphs.** Another used quantum tool is quantum walk search, especially the one over Johnson graphs. Define a search problem related to graph $G$. Let $F$ be a finite set with $|F| = O(n^c)$ for constant $c$, and $r \leq |F|$. We denote the set of all $r$-size subsets of $F$ by $\binom{F}{r}$, which has cardinality $\binom{|F|}{r}$. Given a Boolean function $f_G\colon \binom{F}{r} \to \{0, 1\}$ depending on $G$. The goal is to decide whether $f_G^{-1}(1) \neq \emptyset$. Johnson graph $J(F, r)$ has vertex set $\binom{F}{r}$, where two vertices are adjacent if they differ in exactly one element.

**Lemma 2.2** (Quantum walk search, Ambainis (2007); Magniez et al. (2007a); Bonnetain et al. (2023)). *Suppose $|f_G^{-1}(1)|/\binom{|F|}{r} \geq \varepsilon$ when $f_G^{-1}(1) \neq \emptyset$. Then a marked element can be found with probability at least $3/4$ in time $\widetilde{O}\left(\mathsf{S} + \frac{1}{\sqrt{\varepsilon}}\left(\sqrt{r}\,\mathsf{U} + \mathsf{C}\right)\right)$, where $\mathsf{S}, \mathsf{U}, \mathsf{C}$ denote setup, update, and checking costs.*

More details and useful tools are deferred to Appendix C.

# 3. Quantum Algorithms for Triangle Listing

In this section, we present quantum algorithms for triangle listing based on multiple algorithmic paradigms. Our results combine (i) a heavy–light vertex partition framework for sparse graphs, (ii) extensions of triangle detection via quantum walk search, and (iii) a Grover search approach.

## 3.1. Heavy–Light Vertex Partition

We present the quantum triangle listing algorithm based on heavy–light vertex partition. Let $\Delta$ be a threshold to be specified later, and we partition the vertex set $V$ into heavy vertices $V_H = \{v \in V : \deg(v) > \frac{9}{10}\Delta\}$ and light vertices $V_L = \{v \in V : \deg(v) \leq \frac{11}{10}\Delta\}$, where the overlap accounts for estimation slack (in the case degree needs estimation) and does not affect correctness. Since $\sum_{v \in V} \deg(v) = 2m$, we have $|V_H| \leq \frac{20m}{9\Delta}$. Then every triangle in the graph falls into one of two categories: tri-

---

**Algorithm 1** HL-TRIANGLELISTING($\mathcal{O}_G, \Delta$)

1: $\mathcal{T}_L \leftarrow \emptyset, \mathcal{T}_H \leftarrow \emptyset$
2: Partition $V$ into $V_L$ and $V_H$
3: **for each** $v \in V_L$ **do**
4:     $\mathcal{R}_v = \{\{v, u, w\} : u, w \in N(v)\}$
5:     $\mathcal{T}_L$ add triangles found by repeated Grover search on $\mathcal{R}_v$
6:     Remove $v$ and incident edges from $\mathcal{O}_G$
7: **end for**
8: $\mathcal{T}_H \leftarrow$ triangles found by repeated Grover search on all triples of the remaining graph
9: **return** $\mathcal{T}_L \cup \mathcal{T}_H$

---

angles containing at least one light vertex, and triangles induced entirely by heavy vertices. The algorithm is given in Algorithm 1, with the following guarantee.

**Theorem 3.1.** *Let $G = (V, E)$ be a graph with $n$ vertices, $m$ edges, and $t$ triangles. Setting $\Delta = \sqrt{m}$, HL-TRIANGLELISTING outputs all triangles with high probability in $\widetilde{O}(m + m^{3/4}t^{1/2})$ time.*

*Proof.* We first compute the heavy–light vertex partition by degree queries in time $O(n)$. Every triangle in $G$ either contains a light vertex or is fully contained in $V_H$. Removing each light vertex after processing ensures that every triangle is listed, and exactly once.

**Light Vertices.** Fix $v \in V_L$ and let $t_v$ denote the number of triangles incident to $v$. We enumerate $N(v)$ using neighbor queries in $\widetilde{O}(\deg(v))$ time. The search space $\mathcal{R}_v$ has size $\binom{\deg(v)}{2}$, and repeated Grover search lists all such triangles in time $\widetilde{O}\left(\sqrt{\binom{\deg(v)}{2} t_v}\right) = \widetilde{O}(\deg(v)\sqrt{t_v})$. Summing over all $v \in V_L$ and applying Cauchy–Schwarz inequality, $\sum_{v \in V_L} \deg(v)\sqrt{t_v} \leq \sqrt{\left(\sum_{v \in V_L} \deg^2(v)\right)\left(\sum_{v \in V_L} t_v\right)}$. Each triangle contains three vertices, so $\sum_{v \in V_L} t_v \leq \sum_{v \in V} t_v = 3t$. Moreover, since $\deg(v) \leq \Delta$ for all $v \in V_L$, $\sum_{v \in V_L} \deg(v)^2 \leq \Delta \sum_{v \in V_L} \deg(v) \leq 2m\Delta$. Thus, the total time for processing light vertices is $\widetilde{O}(\sqrt{m\Delta t})$.

**Heavy Vertices.** The number of heavy vertices satisfies $|V_H| \leq \frac{20m}{9\Delta}$. Applying the repeated Grover search on the triples of the remaining graph yields time $\widetilde{O}\left(|V_H|^{3/2} t_H^{1/2}\right) = \widetilde{O}\left(\left(\frac{m}{\Delta}\right)^{3/2} t^{1/2}\right)$, where $t_H \leq t$ is the number of triangles induced entirely by heavy vertices.

Choosing $\Delta$ to balance the two terms, $\sqrt{m\Delta t} = (m/\Delta)^{3/2}\sqrt{t}$, gives $\Delta = \sqrt{m}$. Note that the cost of partitioning is $O(n)$, which is dominated by $m$ by removing isolated vertices in advance, gives the claimed $\widetilde{O}(m + m^{3/4}t^{1/2})$ running time. $\square$

### 3.2. Quantum Walk Approach

Our second approach is based on quantum walk search and generalizes the triangle *finding* framework of Le Gall (2014) to triangle *listing*. The resulting algorithm, QW-TRIANGLELISTING (Algorithm 2), is particularly efficient when the number of triangles is moderate. We present high-level descriptions here and defer technical details and proofs to Appendix D.1. The algorithm proceeds in two phases.

**Pre-listing.** In the PRE-LISTING phase, we sample a random vertex subset $S \subseteq V$ and use repeated Grover search to enumerate all triangles containing at least one vertex in $S$. Let $t_1$ denote the number of such triangles. For each discovered triangle, we additionally list all triangles incident to its edges and then remove these edges from the graph oracle. Let $t_1'$ be the number of additional triangles.

For any vertex set $X \subseteq V$, define $\Delta_G(X, S)$ as the set of vertex-pairs in $X$ that do not share a common neighbor in $S$. Note that for any pair $\{v, w\} \in \binom{X}{2} \setminus \Delta_G(X, S)$, there exists a vertex $u \in S$ such that $\{v, w\} \subseteq N_G(u)$. Consequently, if $\{v, w\} \in E$, then $(u, v, w)$ forms a triangle containing a vertex from $S$; otherwise, $\{v, w\}$ does not participate in any triangle in $G$ at all. After PRE-LISTING, every remaining triangle in the graph is supported entirely on vertex-pairs in $\Delta_G(V, S)$.

**Lemma 3.2.** *Let $G = (V, E)$ be an $n$-vertex graph. With high probability, PRE-LISTING finds all triangles containing at least one vertex from $S$ in time $\widetilde{O}\left(ns^{1/2}t_1^{1/2} + \sqrt{nt_1 t_1'}\right)$. The initialization for $H_0$ of size $h$ requires $O(sh)$ time. Moreover, after the procedure terminates, every remaining triangle in $G$ is supported entirely on vertex-pairs contained in $\Delta_G(V, S)$.*

For set $X \subseteq V$ and a vertex $w \in V$, let $\Delta_G(X, w, S) = \binom{N_G(w)}{2} \cap \Delta_G(X, S)$ denote the set of vertex-pairs in $X$ that do not share any common neighbor in $S$, but do share $w$ as a common neighbor. By the randomness in the choice of $S$, we have such a good property $\sum_{w \in V} |\Delta_G(X, w, S)| \leq \frac{n|X|^2}{s}$ with high probability. Consequently, the total number of candidate triangles vertex-pairs in $X$ that can participate in is reduced from $n|X|^2$ to $\frac{n|X|^2}{s}$. This preprocessing step significantly reduces the search space for the quantum walk search in the second phase.

**Exhaustive Listing.** After the previous step, let $t_2$ denote the number of remaining triangles. The second phase repeatedly applies a two-level quantum walk subroutine, Q-WALKSEARCH, to identify a triangle supported on $\Delta_G(V, S)$. Each time a new triangle is found, all triangles incident to its edges are enumerated, and its three edges are then removed from the graph oracle in INCIDENTTRIANGLELISTING (Algorithm 4). The process terminates once no triangles remain.

We implement edge removal by modifying the *graph oracle* $\mathcal{O}_G$ via standard oracle masking. Specifically, we associate an auxiliary qubit, initialized to 0, with each edge index, and flip it to 1 once the edge is removed (i.e., belongs to $E_I$). The modified oracle returns an edge if and only if it exists in the original graph and its auxiliary qubit remains 0. This technique is standard when extending Grover search from finding a single marked element (Lemma C.1) to listing all marked elements (Lemma C.2), by disabling previously discovered items in the search oracle.

**Two-Level Quantum Walk Search.** We employ a two-level quantum walk to find a marked vertex-pair set $H^*$ (i.e., containing an edge from some remaining triangle). The *outer* walk is over Johnson graph $J(V, h)$, consisting of $h$-size vertex subsets $H \subseteq V$. We know $\Delta_G(H, S)$ contains all vertex-pairs that may still support a triangle after PRE-LISTING.

To check whether $\Delta_G(H, S)$ contains an edge that participates in a triangle, we search for a vertex $w \in V$ such that $w$ completes a triangle with some pair in $\Delta_G(H, S)$. For a fixed $w$, this task is handled by an *inner* quantum walk over Johnson graph $J(H, h^{2/3})$ (whose vertices are smaller subsets $H' \subseteq H$), which checks whether $\Delta_G(H', w, S)$ contains an actual edge of $G$. If such a pair exists, it forms a triangle together with $w$.

The analysis differs from triangle detection mainly by bounding the fraction of marked states in the outer walk. Unlike the simple edge-counting argument used in prior work, we bound the marked fraction via a combinatorial packing argument: we take a maximum collection of pairwise edge-disjoint remaining triangles, extract a set of certifying edges of low maximum degree, and apply a second-moment calculation. This yields a rigorous lower bound on the marked fraction that does not assume independence of triangle-supporting edges. The following lemma summarizes the resulting search cost.

**Lemma 3.3.** *Given the graph remaining after* PRE-LISTING *with $n$ vertices and $t_2$ remaining triangles, let $\nu$ be the maximum number of pairwise edge-disjoint triangles in this graph. There exists a quantum walk search algorithm on the Johnson graph $J(V, h)$, starting from an initial state $H_0$, that, with high probability, finds a set $H \subseteq V$ and a vertex $w \in V$ such that $\Delta_G(H, w, S) \cap E \neq \emptyset$. Denote the found set and vertex by $H^*$ and $w^*$. The running time of the algorithm is*

$$\widetilde{O}\Big(\big(1 + \frac{n}{h\sqrt{\nu}}\big) \cdot \big(sh^{1/2} + n^{1/2}h^{2/3} + n^{1/2}s^{-1/2}h\big)\Big),$$

*where $s$ is the sample size in* PRE-LISTING.

**Setting Parameters.** For the parameter setting below, assume for the moment that QW-TRIANGLELISTING is given

---

**Algorithm 2** QW-TRIANGLELISTING($\mathcal{O}_G, n, \hat{t}$)

1: Set parameters $s, h$ according to the estimate $\hat{t}$
2: $\mathcal{T}(G), H_0 \leftarrow$ PRE-LISTING($\mathcal{O}_G, n, s, h$)
3: **while** true **do**
4:    $T \leftarrow$ Q-WALKSEARCH($\mathcal{O}_G, H_0, h$)
5:    **if** $T = \emptyset$ **then**
6:       **return** $\mathcal{T}(G)$
7:    **end if**
8:    Let $E_T$ be the three edges of triangle $T$
9:    $\mathcal{T}' \leftarrow$ INCIDENTTRIANGLELISTING($\mathcal{O}_G, E_T$)
10:   $\mathcal{T}(G) \leftarrow \mathcal{T}(G) \cup \{T\} \cup \mathcal{T}'$
11: **end while**

---

**Algorithm 3** PRE-LISTING($\mathcal{O}_G, n, s, h$)

1: Sample a set $S \subseteq V$ of size $\Theta(s \log n)$ uniformly at random
2: $\mathcal{R}_S = \{\{u, v, w\} \mid u \in S, \{v, w\} \in \binom{V}{2}\}$
3: $\mathcal{T}_S \leftarrow$ triangles found by repeated Grover search on $\mathcal{R}_S$
4: $E_S = \{\{u, v\} \in E : \exists T \in \mathcal{T}_S, \{u, v\} \subseteq T\}$
5: $\mathcal{T}'_S \leftarrow$ INCIDENTTRIANGLELISTING($\mathcal{O}_G, E_S$)
6: Initialize a quantum walk state $H_0$ with $|H_0| = h$
7: **return** $\mathcal{T}_S \cup \mathcal{T}'_S, H_0$

---

an estimate $\hat{t} = \Theta(t)$ of the total number of triangles; this assumption will be removed later by a constant-ratio guessing schedule. No explicit knowledge of $t_1$, $t'_1$ or $t_2$ is needed. Based on $\hat{t}$, we choose parameters $(s, h)$ to balance the costs of PRE-LISTING and quantum walk search phase.

$$(s, h) = \begin{cases} \big(n^{1/2}\hat{t}^{1/6}, \ n^{3/4}\hat{t}^{1/4}\big), & \hat{t} \leq n^{3/7}, \\ \big(n^{1/2}\hat{t}^{1/6}, \ n\hat{t}^{-1/3}\big), & n^{3/7} < \hat{t} \leq n^{3/5}, \\ \big(n^{2/3}\hat{t}^{-1/9}, \ n\hat{t}^{-1/3}\big), & \hat{t} > n^{3/5}. \end{cases}$$

With these choices, we obtain the following guarantee.

**Theorem 3.4.** *Let $G$ be an undirected graph with $n$ vertices and $t$ triangles. There exists a quantum algorithm,* QW-TRIANGLELISTING, *with high probability, that outputs all triangles in $\widetilde{O}\big(n^{5/4}t^{7/12} + n^{7/6}t^{7/9}\big)$ time.*

We defer the Grover-search-based algorithm and its analysis to Appendix D.2.

**Putting Things Together.** We combine the triangle listing algorithms based on heavy–light vertex partition, quantum walk, and Grover search to obtain the final bound.

*Proof of Theorem 1.3.* Q-TRIANGLELISTING run the algorithms of Theorem 3.1, Theorem 3.4, and Theorem D.10 in parallel, and return the output from the one that terminates first. The resulting running time is $\widetilde{O}\big(\min(n^{5/4}t^{7/12} + n^{7/6}t^{7/9}, m + m^{3/4}t^{1/2}, n^{3/2}t^{1/2})\big)$. □

---

**Algorithm 4** INCIDENTTRIANGLELISTING($\mathcal{O}_G, E_I$)

---

1: $\mathcal{R}_E = \{\{u, v, w\} \mid u \in V, \{v, w\} \in E_I\}$
2: $\mathcal{T} \leftarrow$ triangles found by repeated Grover search on $\mathcal{R}_E$
3: Modify $\mathcal{O}_G$ by removing all edges in $E_I$
4: **return** $\mathcal{T}$

---

**Algorithm 5** Q-WALKSEARCH($\mathcal{O}_G, H_0, h$)

---

1: Run the two-level quantum walk starting from $H_0$ (Lemma 3.3)
2: Obtain a marked vertex $w^*$ and a subset $H^* \subseteq V$
3: $\mathcal{R}_{H^*} = \{\{w^*, u, v\} \mid \{u, v\} \in \binom{H^*}{2}\}$
4: Use standard Grover search on $\mathcal{R}_{H^*}$
5: **return** the triangle found, or $\emptyset$ if none exists

---

## 4. Quantum Algorithms for Triangle Cut Sparsification

We now present the quantum algorithm for triangle cut sparsification that leverages our triangle listing algorithm. Note that directly applying quantum hypergraph sparsification, by representing each triangle as a hyperedge, does not yield what we want, since an edge can appear in both retained and discarded hyperedges at the same time. Instead, we base our quantum algorithm on the classical strength-based motif cut sparsification framework of Kapralov et al. (2022). For completeness, we provide the full classical algorithm, and its time analysis in Appendix E.

Once all the triangles are listed, the algorithm iteratively sparsifies the graph. Each iteration of the algorithm consists of two steps: finding critical edges and post-sampling the remaining edges. We show that both steps admit efficient quantum implementations, leading to an overall quantum speedup for triangle cut sparsification.

### 4.1. Finding Critical Edges

We accelerate the first step by combining quantum triangle listing with Grover search. As in the classical construction, this step identifies edges whose total importance exceeds a prescribed threshold. We begin with the formal definition about triangle strength from Kapralov et al. (2022).

**Definition 4.1** (Triangle Connectivity). Let $G = (V, E, w)$ be an undirected weighted graph. We say that $G$ is $k$-connected (with respect to triangles) if every cut $(S, V \setminus S)$ in $G$ has triangle size at least $k$.

**Definition 4.2** ($k$-Strongly Connected Component). Let $G = (V, E, w)$ be an undirected weighted graph. For a value $k \in \mathbb{R}_+$, an induced subgraph $C = (V_C, E_C, w)$ of $G$ is called a $k$-strongly connected component of $G$ if

(i) $C$ is $k$-connected, and
(ii) there is no induced subgraph $C' = (V_{C'}, E_{C'}, w)$ of

$G$ that is $k$-connected and satisfies $V_C \subsetneq V_{C'}$.

**Definition 4.3** (Triangle Strength). Let $G = (V, E, w)$ be an undirected weighted graph and let $\mathcal{T}(G)$ denote the set of all triangle instances in $G$. For a triangle instance $T \in \mathcal{T}(G)$, the triangle strength $\kappa_T$ is the maximum value $k$ such that there exists a $k$-strongly connected component that contains $T$ as a subgraph.

Due to the expensive computation, we only approximate importance weight of each edge in the algorithm, using existed hypergraph strength estimation method (Chekuri & Xu, 2018) to obtain the strength estimate $\kappa'_T$ of triangle $T$.

---

**Algorithm 6** Q-CRITICALEDGEFINDING($\mathcal{O}_G, \varepsilon$)

---

1: $\mathcal{T}(G) \leftarrow$ Q-TRIANGLELISTING($\mathcal{O}_G$)
2: $\{\kappa'_T\}_{T \in \mathcal{T}(G)} \leftarrow$ STRENGTHESTIMATION($\mathcal{O}_G, \mathcal{T}(G)$)
3: $E_+ \leftarrow$ critical edges found by repeated Grover search on $E$
4: **return** $E_+$

---

**Lemma 4.4.** *Let $G = (V, E, w)$ be an undirected weighted graph. Given query access $\mathcal{O}_G$ to the graph, Q-CRITICALEDGEFINDING outputs the critical edge set $E_+$ in time $T_{\text{q-list}} + \widetilde{O}(|\mathcal{T}(G)| + \sqrt{m|E_+|})$ with high probability, where $m = |E|$ and $T_{\text{q-list}}$ denotes the running time of quantum triangle listing.*

*Proof.* We list all triangles $\mathcal{T}(G)$ in time $T_{\text{q-list}}$ with high probability by Theorem 1.3, and then compute an estimated strength for each triangle via the classical routine STRENGTHESTIMATION (Chekuri & Xu, 2018), obtaining $\{\kappa'_T\}_{T \in \mathcal{T}(G)}$. By Lemma E.1, it runs in $\widetilde{O}(|\mathcal{T}(G)|)$ time. These estimates determine the importance weight contributed by each triangle.

All intermediate data, including weights and estimated strengths, are stored in QRAM. We thus assume QRAM access to $\mathcal{T}(G)$ and the associated values. Using this access, we precompute the estimated edge importance $\widehat{\eta}(e)$ for every edge $e \in E$ as follows:

1. Initialize $\widehat{\eta}(e) \leftarrow 0$ for all $e \in E$.
2. For each triangle $T \in \mathcal{T}(G)$, add $w(T)/\kappa'_T$ to $\widehat{\eta}(e)$ for all three edges $e$ of $T$.

This preprocessing step touches each triangle three times and thus costs $\widetilde{O}(|\mathcal{T}(G)|)$ time, which is dominated by the earlier triangle listing phase. After this preprocessing, the importance $\widehat{\eta}(e)$ of any edge $e$ can be queried in $\widetilde{O}(1)$ quantum time via QRAM read. Let $c_1, d$ be absolute constants and set $\varepsilon' = \varepsilon/(15c_1 \log n)$. Define the Boolean predicate $g(e) = 1$ if $\widehat{\eta}(e) > \frac{d\varepsilon'^2}{3(\log n + 3)}$; and 0 otherwise. It decides whether $e$ is a critical edge. Hence, for any edge $e$, we can evaluate the predicate $g$ in $\widetilde{O}(1)$ time.

Applying repeated Grover search (Lemma 2.1) over $E$ with predicate $g$ finds all critical edges in time $\widetilde{O}(\sqrt{m|E_+|})$. Summing all costs yields the stated running time. $\qquad\square$

### 4.2. Post-Sampling the Remaining Edges

We now describe how to accelerate the post-sampling step. Explicitly outputting all edges in the graph after each iteration would require up to $O(m)$ time since the expected number of surviving edges is in this order. To resolve this issue, we follow the techniques of Apers & De Wolf (2022) and implement sampling implicitly using random strings, which allows us to defer outputting the sparsified graph.

**Implicit Sampling via Random Strings.** Assume access to a family of random strings $\{r_i\}_{1 \le i \le \lceil 6c_1 \log n \rceil}$, where each $r_i$ is a binary string of length $m$. All bits of each $r_i$ are independent and equal to $0$ with probability $1 - p = 1 - 2^{-1/6}$. Each bit of $r_i$ corresponds to an edge in $E$.

In iteration $i$, after computing the critical edge set $E_+$, we sample each non-critical edge $e \in E \setminus E_+$ according to the random bit. We set $w'(e) = 0$ if $r_i(e) = 0$; or $w'(e) = w(e)/p$ if $r_i(e) = 1$. After all iterations, apply repeated Grover search to identify edges with nonzero weight and these edges constitute the final sparsifier.

---

**Algorithm 7** Q-TRIANGLECUTSPARSIFICATION($\mathcal{O}_G, \varepsilon$)

---

1: **for** $i = 1$ **to** $\lceil 6c_1 \log n \rceil$ **do**
2: $\quad E_+ \leftarrow$ Q-CRITICALEDGEFINDING($\mathcal{O}_G, \frac{\varepsilon}{15c_1 \log n}$)
3: $\quad$ **for** each edge $e \in E \setminus E_+$ **do**
4: $\quad\quad$ **if** $r_i(e) = 0$ **then**
5: $\quad\quad\quad w'(e) \leftarrow 0$
6: $\quad\quad$ **else**
7: $\quad\quad\quad w'(e) \leftarrow w(e)/p$
8: $\quad\quad$ **end if**
9: $\quad$ **end for**
10: $\quad w(e) \leftarrow w'(e)$
11: **end for**
12: $E' = \{e \in E \mid w'(e) > 0\}$ found by repeated Grover search
13: **return** $G' = (V, E', w')$

---

**Theorem 4.5.** *Given oracle access to independent, uniformly random strings $r_i \in \{0,1\}^m$ for $1 \le i \le \lceil 6c_1 \log n \rceil$, Q-TRIANGLECUTSPARSIFICATION outputs, with high probability, an $\varepsilon$-triangle cut sparsifier of $G$ with $\widetilde{O}(n/\varepsilon^2)$ edges. Moreover, there exists a quantum implementation with running time $\widetilde{O}(T_{\text{q-list}} + \sqrt{mn}/\varepsilon)$.*

*Proof.* Correctness follows directly from Theorem E.4. In each iteration, the algorithm partially sparsifies the current graph induced by edges with nonzero weight. By a union bound over all $O(\log n)$ iterations, the probability that all

iterations succeed is $1 - O(\log n / \operatorname{poly}(n))$.

We now analyze the running time. The dominant cost per iteration is identifying the critical edges, which by Lemma 4.4 takes $T_{\text{q-list}} + \widetilde{O}(\sqrt{m|E_+|})$ time (note that $T_{\text{q-list}}$ is always no less than the total number of triangles $|\mathcal{T}(G)|$). Since the size of critical edges $|E_+| = \widetilde{O}(n/\varepsilon^2)$ by Lemma E.5, the total cost over all iterations is $\widetilde{O}(T_{\text{q-list}} + \sqrt{mn}/\varepsilon)$. Indeed, we can compute $\mathcal{T}(G)$ only once, since we only delete edges during the algorithm. In the final step, finding $\widetilde{O}(n/\varepsilon^2)$ preserved edges of the sparsifier (Lemma E.6) among $m$ edges requires $\widetilde{O}(\sqrt{mn}/\varepsilon)$ time.

It remains to show that the edge sampling and reweighting during each iteration can be implemented efficiently. Weights are not updated explicitly. Instead, for each edge $e$, the algorithm maintains an implicit representation of its final weight using the random strings and the history of critical-edge selections. Consider the $i$-th iteration. Let $k$ denote the number of times edge $e$ has been selected as a critical edge prior to iteration $i$. If $k = 0$, then

$$w'(e) = \begin{cases} (\frac{1}{p})^i w(e), & \text{if } (r_i r_{i-1} \ldots r_1)(e) = 1, \\ 0, & \text{otherwise.} \end{cases}$$

If $k > 0$, let $i' < i$ be the most recent iteration in which $e$ was selected as a critical edge. Then

$$w'(e) = \begin{cases} (\frac{1}{p})^{i-k} w(e), & \text{if } (r_i r_{i-1} \ldots r_{i'+1})(e) = 1, \\ 0, & \text{otherwise.} \end{cases}$$

This logic can be implemented by an oracle that computes $w'(e)$ on demand and is used within Grover search. $\qquad\square$

**Removing Random Strings.** Finally, we eliminate the assumption of access to random strings. The following lemma gives the guarantee, with at most a polylogarithmic overhead in the runtime.

**Lemma 4.6** (Corollary 3.3 in (Apers & De Wolf, 2022)). *Any quantum algorithm with runtime $q$ that makes queries to a uniformly random string can be simulated by a quantum algorithm with runtime $\widetilde{O}(q)$ without random string, using an additional QRAM of $\widetilde{O}(q)$ bits.*

*Proof of Theorem 1.2.* The theorem follows immediately by combining Theorem 4.5 with Lemma 4.6. $\qquad\square$

## 5. Extensions

### 5.1. Applications: Triangle Clustering

Triangle clustering captures higher-order community structure by measuring cluster quality using *triangle-based conductance* (see Definition F.2) rather than edges alone, and has been shown to reveal substantially different partitions

in practice and theory. A key structural fact underlying existing algorithms is that triangle clustering reduces exactly to standard conductance on a *triangle-weighted graph*, where each edge weight equals the total weight of triangles containing that edge. As a result, classical triangle clustering algorithms first construct this triangle-weighted graph and then apply standard spectral or PageRank-based clustering procedures (Austin R. Benson & Leskovec, 2016; Tsourakakis et al., 2017; Yin et al., 2017), inheriting the corresponding approximation guarantees.

Our quantum algorithms yield a natural speedup of this pipeline by accelerating its dominant computational bottleneck—triangle enumeration. Using our quantum triangle listing algorithm, we can implement query access to the triangle-weighted graph more efficiently than classical methods, which suffices for downstream clustering algorithms. This allows us to combine our method with existing quantum primitives such as quantum spectral clustering (Apers & De Wolf, 2022), yielding quantum speedups for both global (2-way and $k$-way) triangle spectral clustering and local triangle clustering, while preserving the same theoretical guarantees as their classical counterparts. For example, we have the following corollary.

**Corollary 5.1.** *There exists a quantum algorithm that, given query access to G, returns a 2-partition in time $T_{q\text{-list}} + \widetilde{O}(\sqrt{mn})$. The triangle conductance of the output clustering is at most $4\sqrt{\phi_2^{\Delta}(G)}$.*

Full definitions, formal reductions, and detailed guarantees are provided in Appendix F.

### 5.2. Lower Bound on the Size of Triangle Cut Sparsifiers

We show that the sparsity achieved by our triangle cut sparsification is essentially optimal.

**Theorem 5.2.** *For every integer n and $\varepsilon \in (1/\sqrt{n}, 1)$, there exists an n-vertex graph G such that every $\varepsilon$-triangle cut sparsifier of G has $\Omega(n/\varepsilon^2)$ edges.*

The proof proceeds via a reduction from the one-way distributional Gap-Hamming-Distance problem, inspired by the classical construction in (Andoni et al., 2016). We lift their bipartite hard instance of cut sparsifiers to a triangle-weighted graph, and show that triangle cut values in the original graph correspond, up to a constant factor, to cut values in the associated triangle-weighted graph. Consequently, a triangle cut sparsifier with fewer than $n/\varepsilon^2$ edges would yield a protocol violating the communication lower bound. The full construction and proof are deferred to Appendix G.

Together with our upper bounds, Theorem 5.2 implies that our $\widetilde{O}(n/\varepsilon^2)$-size triangle cut sparsifiers are nearly tight up to polylogarithmic factors. In particular, since our sparsifier size inherits that of Kapralov et al. (2022), this result

establishes near-optimality of their sparsity guarantees for triangle motifs. We note that existing lower bounds in that work apply to induced-motif cut sparsifiers (i.e., 2-paths), whereas our result provides a matching lower bound specifically for triangle cut sparsification.

## 6. Future Work

We raise several open questions for future work.

**General Motifs.** A natural question is whether the quantum speedup for triangle cut sparsification can be extended to more general motifs. The post-sampling and reweighting stage of our framework is motif-agnostic; the main bottleneck lies in the motif listing primitive. Our quantum triangle listing algorithm exploits structural properties specific to triangles—for instance, that a triangle is determined by a vertex together with a pair of its neighbor, which enables the efficient use of heavy–light decompositions. For larger patterns such as 4-cycles or higher-order motifs, analogous structural primitives are less direct, and obtaining similar polynomial speedups remains an open challenge. Concretely, classical algorithms for listing 4-cycles either enumerate all length-2 paths (leading to $O(n^2 + t)$ time, where $t$ is the number of 4-cycles) or achieve $O(m^{4/3} + t)$ via heavy–light vertex partitioning (Abboud et al., 2022). While our heavy–light techniques share some intuition with the latter approach, the adaptation is nontrivial, and a quantum speedup for 4-cycle listing is left for future study.

**Query Lower Bounds.** Establishing quantum query lower bounds for triangle listing and triangle cut sparsification remains an important direction. For triangle finding, an $\Omega(n)$ quantum lower bound is known via reduction to the OR problem, but it is not tight against the $O(n^{5/4})$ upper bound (Le Gall, 2014). For triangle listing, no nontrivial quantum lower bound is currently known. Existing adversary and polynomial methods are primarily designed for Boolean decision problems, while triangle listing is a global structure recovery problem with non-constant output size $\Theta(t)$, which limits their direct applicability. For (edge) cut sparsifiers, a quantum query lower bound of $\Omega(\sqrt{mn}/\varepsilon)$ has been established (Apers & De Wolf, 2022). It is proved by reducing sparsification to a structured search problem: a large fraction of edges from some unsparsifiable graph must be found within the whole graph. This search task is then formulated as a composition of finding marked entries in a Boolean matrix with the OR function, and the final bound follows by applying a composition theorem. Our $\Omega(n/\varepsilon^2)$ sparsifier-size lower bound, along with the existing quantum sparsification lower-bound framework, may suggest $\Omega(\sqrt{mn}/\varepsilon)$ as a natural baseline query lower bound. Nevertheless, triangle cut sparsification also requires triangle-level information such as triangle counting or listing, and therefore its full quantum query complexity may be larger.

## Acknowledgements

The work is supported in part by Quantum Science and Technology - National Science and Technology Major Project (Grant No. 2021ZD0302901) and NSFC Grant 62272431.

## Impact Statement

This paper presents work whose goal is to advance the field of Machine Learning. There are many potential societal consequences of our work, none which we feel must be specifically highlighted here.

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

# A. Overview of the Appendix

The appendix is organized as follows:

- Appendix B: we provide the discussion of other related work.
- Appendix C: we provide the supplementary tools used in the appendix.
- Appendix D: we present the omitted algorithms and analysis in Section 3.
- Appendix E: we present the deferred algorithms, proofs and analysis from Section 4.
- Appendix F: we present the application on the quantum algorithms for triangle-based clustering.
- Appendix G: we present the lower bound on the size of triangle cut sparsifier.

# B. Other Related Work

**Triangles and Related Problems.** Triangles are fundamental higher-order structures in graphs, capturing interactions beyond pairwise relationships. For instance, the well-known tendency for "friends of friends to become friends" in social networks is naturally modeled through triangles (Wasserman, 1994). Triangles also underpin classical notions such as clustering coefficients (Soffer & Vazquez, 2005; Arifuzzaman et al., 2012) and often appear as building blocks in real-world networks (Milo et al., 2002; Chu & Cheng, 2011; Chen et al., 2020b).

Motivated by these applications, a variety of triangle-related algorithmic primitives have been studied, including triangle-motif conductance (Austin R. Benson & Leskovec, 2016; Tsourakakis et al., 2017) and triangle-based embeddings (Nassar et al., 2020). Triangle listing has also been investigated in distributed models, though almost exclusively in the classical setting; representative results include Chu & Cheng (2011); Giechaskiel et al. (2015); Ha-Myung Park & Kang (2016).

**Quantum Advances.** Quantum algorithms for triangle-related problems have attracted attention also in distributed settings. Izumi et al. (2020) presented a quantum algorithm for triangle finding that runs in $\widetilde{O}(n^{1/4})$ rounds in the CONGEST model. Subsequently, Censor-Hillel et al. (2022) introduced a quantum distributed framework for clique detection, improving the state of the art for triangles and extending to larger cliques.

More broadly, quantum speedups for graph algorithms have been developed through the sparsification paradigm. The foundational work of Apers & De Wolf (2022) initiated quantum algorithms for a series of related problems, along with quantum query lower bounds. Building on this line of work, Apers & Lee (2021) gave a quantum algorithm for exact minimum cut under bounded weight ratios, and Apers et al. (2021) extended these techniques to exact minimum $s$–$t$ cut computation. Subsequent advances include quantum motif clustering (Cade et al., 2023) and a general framework for quantum spectral approximation (Apers & Gribling, 2023). In the hypergraph setting, Liu et al. (2025) developed quantum algorithms for cut sparsification and minimum cut problems.

Besides, quantum algorithms have been applied to a broad range of graph-theoretic problems (Dürr et al., 2006; Magniez et al., 2007b; Lee et al., 2021; Ponce et al., 2025), including minimum spanning tree (O'Quinn & Mao, 2020), random spanning tree sampling (Apers et al., 2025), and graph coloring (Shimizu & Mori, 2022).

# C. Omitted Preliminaries from Section 2

This section collects additional details on the computational model and quantum tools used throughout the paper.

### C.1. Quantum Computing and Computational Model

We refer to Section 2 for a concise introduction to quantum states, unitaries, and measurements.

Our computational model assumes the input graph $G = (V, E, w)$ is stored in a quantum-accessible classical memory (QRAM) that supports the following queries coherently:

- **degree query:** given $v \in V$, return its degree $\deg(v)$;
- **neighbor query:** given $v \in V$ and an index $i \in [\deg(v)]$, return the $i$-th neighbor of $v$;
- **vertex-pair query:** given a pair of vertices $\{u, v\}$, return the weight $w(u, v)$ if $\{u, v\} \in E$, and 0 otherwise.

The weight of an edge $e$ is encoded as floating-point numbers $|w(e)\rangle \in \mathbb{C}^{d_{\mathrm{acc}}}$, where $d_{\mathrm{acc}} = O(1)$ suffices for arbitrary

constant precision. These three queries can be implemented as the following quantum oracles:

$$\mathcal{O}_{\deg} |v\rangle |0\rangle = |v\rangle |\deg(v)\rangle ,$$
$$\mathcal{O}_{\mathrm{neigh}} |v\rangle |i\rangle |0\rangle = |v\rangle |i\rangle |N(v)_i\rangle ,$$
$$\mathcal{O}_{\mathrm{edge}} |\{u,v\}\rangle |0\rangle = \begin{cases} |\{u,v\}\rangle |w(u,v)\rangle , & \text{if } \{u,v\} \in E, \\ |\{u,v\}\rangle |0\rangle , & \text{otherwise.} \end{cases}$$

In analogy with classical RAM, a QRAM is a device that allows storage or modification of its data. Specifically, we can either classically write a bit to QRAM or "quantumly" read bits stored in QRAM, possibly in superposition. As in prior work on quantum algorithms for graph problems (e.g., Apers & De Wolf (2022); Chen et al. (2025b)), we assume the QRAM model to allow quantum query access. While the physical realizability of large-scale QRAM device remains an open question, this assumption is standard and common in the literature and allows us to focus on algorithmic aspects.

### C.2. Quantum Algorithmic Tools

**Quantum Search.** A fundamental quantum subroutine frequently used in algorithm design is Grover search (Grover, 1996). Consider a Boolean predicate: $g_G \colon [N] \to \{0,1\}$ depending on graph $G$ and let $M = \{i \in [N] \mid g_G(i) = 1\}$ denote the marked set. When $M \neq \emptyset$, we have the following lemmas.

**Lemma C.1** (Standard Grover search, Grover (1996)). *There exists a quantum algorithm that finds an element in $M$ with probability at least $2/3$ in $\widetilde{O}(\sqrt{N/|M|})$ elementary operations and queries to $g_G$.*

**Lemma C.2** (Repeated Grover search, stated in (Apers & De Wolf, 2022)). *There exists a quantum algorithm that finds $M$ with probability at least $2/3$ in $\widetilde{O}(\sqrt{N\,|M|})$ elementary operations and queries to $g_G$.*

To derive the latter lemma, we record a list of all found solutions and modify the oracle such that in Grover's algorithm, it flips the phase of $|x\rangle$ if and only if $x$ is in $M$ and $x$ is not in the list yet. This way, we can repeatedly apply Lemma C.1 and thus the complexity for finding all marked elements is $|M| \cdot \widetilde{O}(\sqrt{N/|M|}) = \widetilde{O}(\sqrt{N\,|M|})$.

**Variable-Cost Quantum Search.** Suppose that for each $i \in [N]$, there exists a quantum subroutine $\mathcal{A}_i$ which uses at most $q_i$ queries to $g_G$ and outputs $g_G(i)$ with probability at least $2/3$. Then we have a variable-cost generalization of quantum search.

**Lemma C.3** (Variable-cost quantum search, Ambainis (2010)). *Assume there exists a constant $c$ such that $N \leq n^c$ and $\max_i\{q_i\} \leq n^c$. Then there is a quantum algorithm that finds an element in $M$ with probability at least $3/4$ using $\widetilde{O}\left(\sqrt{\sum_{i \in [N]} q_i^2}\right)$ queries.*

**Quantum Walk Search.** Another important quantum tool is quantum walk search. We specifically focus on quantum walks over Johnson graphs, as they suffice for our technical purposes. See (Magniez et al., 2007a) for a more comprehensive and general treatment of quantum walks. We denote the set of all $r$-size subsets of $F$ by $\binom{F}{r}$, which has cardinality $\binom{|F|}{r}$. We then define the Johnson graph $J(F,r)$ as follows.

**Definition C.4** (Johnson graph). *Let $F$ be a finite set and $r$ be a positive integer such that $r \leq |F|$. The Johnson graph $J(F,r)$ is an undirected graph with node set $\binom{F}{r}$, where two nodes $R_1, R_2 \in \binom{F}{r}$ are connected if and only if $|R_1 \cap R_2| = r - 1$.*

We describe a search problem related to graph $G$ (accessed via oracle $\mathcal{O}_G$). Let $F$ be a finite set with $|F| = O(n^c)$ for constant $c$, and $r \leq |F|$. Given a Boolean function $f_G \colon \binom{F}{r} \to \{0,1\}$ depending on $G$, define $M_G := f_G^{-1}(1)$. The goal is to decide whether $M_G \neq \emptyset$.

This problem can be solved using a quantum walk over the Johnson graph $J(F,r)$. Each walk state corresponds to a node (i.e., a set $A \in \binom{F}{r}$) and its associated data structure $D_G(A)$ related with $G$. A state (or, a set) $A$ is marked when $A \in M_G$. Three types of cost are associated with $D_G$: the setup cost $\mathsf{S}$, to construct $D_G(A)$ for a given node $A \in \binom{F}{r}$; the update cost $\mathsf{U}$, to convert $D_G(A)$ into $D_G(A')$ for two given connected nodes $A$ and $A'$ in $J(F,r)$; the checking cost $\mathsf{C}$, to check with probability greater than $2/3$ whether the state $A$ is marked for the given $A \in \binom{F}{r}$ and $D_G(A)$. We have the lemma.

**Lemma C.5** (Quantum walk search, Ambainis (2007); Magniez et al. (2007a); Bonnetain et al. (2023)). *Let $\frac{|M_G|}{\binom{|F|}{r}} \geq \varepsilon > 0$ whenever $M_G \neq \emptyset$. Suppose that $M_G \neq \emptyset$, then the quantum walk over the Johnson graph $J(F,r)$ finds an element in $M_G$, with probability at least $3/4$, using $\widetilde{O}\left(\mathsf{S} + \frac{1}{\sqrt{\varepsilon}}\left(\sqrt{r} \times \mathsf{U} + \mathsf{C}\right)\right)$ elementary operations and queries to $\mathcal{O}_G$.*

Besides, we give a fact that relates the number of triangles and edges participating in these triangles.

**Fact C.6.** *Any graph with $m$ edges contains at most $\frac{\sqrt{2}}{3} m^{3/2}$ triangles.*

*Proof.* Let $G$ be a graph with $m$ edges and $t$ triangles. For each vertex $v_i$, let $N_i$ be its neighborhood and let $E_i$ denote the set of edges with both endpoints from $N_i$. Then the number of triangles containing $v_i$ is exactly $|E_i|$. Clearly, $|E_i| \leq m$, and also $|E_i| \leq \binom{|N_i|}{2} \leq \frac{|N_i|^2}{2}$. Combining these inequalities yields $|E_i|^2 \leq \frac{m}{2}|N_i|^2$, hence $|E_i| \leq \sqrt{m/2}\,|N_i|$. Summing over all vertices yields $3t = \sum_i |E_i|$, so $3t \leq \sum_i \sqrt{\frac{m}{2}}|N_i| = \sqrt{\frac{m}{2}} \sum_i |N_i| = \sqrt{\frac{m}{2}} \cdot 2m = \sqrt{2}m^{3/2}$. $\square$

# D. Omitted Algorithms and Analysis in Section 3

This appendix contains the algorithms, proofs, and technical details that were omitted from Section 3. For completeness and ease of reference, we restate the relevant results before presenting their detailed analysis.

## D.1. Analysis of QW-TRIANGLELISTING

In this section, we analyze the performance of our quantum triangle listing algorithm, QW-TRIANGLELISTING. We begin with the analysis of the frequently used subroutine INCIDENTTRIANGLELISTING, which finds all triangles incident to a given edge set $E_I$ and subsequently removes them from the graph. We then analyze the step PRE-LISTING, where a vertex subset of size $\Theta(s \log n)$ is sampled. Finally, we turn to the quantum walk search subroutine Q-WALKSEARCH, and conclude the section with the statement and proof of our main theorem.

### D.1.1. ANALYSIS OF INCIDENTTRIANGLELISTING

We first analyze the subroutine INCIDENTTRIANGLELISTING (Algorithm 4). We say that a triangle $T$ is *incident* to a set of edges $E' \subseteq E$ if $T$ contains at least one edge in $E'$. Recall that in the invocation of INCIDENTTRIANGLELISTING, the set $E_I$ consists only of edges that have already been identified as participating in at least one triangle.

**Lemma D.1.** *Let $G = (V, E)$ be an $n$-vertex graph. Given a set $E_I \subseteq E$ such that every edge in $E_I$ is contained in at least one triangle of $G$, INCIDENTTRIANGLELISTING finds all triangles incident to $E_I$ with high probability in time $\widetilde{O}\left(\sqrt{n|E_I| \cdot t_{E_I}}\right)$, where $t_{E_I}$ denotes the number of triangles incident to $E_I$.*

*Proof.* To support quantum search, we define a Boolean function $f \colon V^3 \to \{0, 1\}$ by

$$f(u, v, w) = \begin{cases} 1, & \text{if } (u, v), (v, w), (w, u) \in E \\ 0, & \text{otherwise.} \end{cases}$$

Quantum access to $f$ is provided via an oracle $\mathcal{O}_f$. Given a triple $(u, v, w)$, $\mathcal{O}_f$ can be implemented using three queries to the graph oracle $\mathcal{O}_G$ to check the existence of the three edges, followed by $\widetilde{O}(1)$ additional elementary operations. Hence, each evaluation of $\mathcal{O}_f$ has $\widetilde{O}(1)$ overhead.

The size of search space in INCIDENTTRIANGLELISTING is $|\mathcal{R}_E| = n|E_I|$, and the number of marked elements is exactly $t_{E_I}$ (where a marked element corresponds to a triangle incident to $E_I$). Applying the repeated Grover search (Lemma C.2), we can list all marked elements in $\widetilde{O}\left(\sqrt{n|E_I| \cdot t_{E_I}}\right)$ time with constant probability. Repeat this process $O(\log n)$ times and take their union to boost the success probability.

Once all such triangles are identified, the edges in $E_I$ can be safely removed without missing any remaining triangles. This prevents duplicate listings arising from overlapping triangles. We then show how to modify the *graph oracle* $\mathcal{O}_G$ to remove edges—this modification on $\mathcal{O}_G$ can be implemented by associating an auxiliary qubit (initialized to 0) with each edge index, and when the search finishes, setting it to 1 if the corresponding edge is in $E_I$. Thereafter, the modified oracle returns an edge only if it exists in the original graph and the auxiliary qubit is set to 0. Such oracle masking is standard when extending Grover search from finding a single marked element (Lemma C.1) to listing all marked elements (Lemma C.2), by disabling every found element in the search oracle.

Finally, we account for the overhead of oracle masking. Since every edge in $E_I$ participates in at least one triangle, and each triangle contains exactly three edges, we have the combinatorial bound $3t_{E_I} \geq |E_I|$. Therefore, the total cost of masking all edges in $E_I$ is $\widetilde{O}(|E_I|)$, which is dominated by the Grover search complexity. This completes the proof. $\square$

D.1.2. ANALYSIS OF PRE-LISTING

We now analyze the procedure PRE-LISTING (Algorithm 3). For any vertex sets $S, X \subseteq V$, we define a set of vertex-pairs $\Delta_G(X, S) \subseteq \binom{X}{2}$ as follows:

$$\Delta_G(X, S) = \binom{X}{2} \setminus \bigcup_{u \in S} \binom{N_G(u)}{2},$$

where $N_G(u)$ is the neighborhood of $u$ in $G$. Namely, $\Delta_G(X, S)$ consists of all vertex-pairs in $X$ that do not share any common neighbor in $S$.

Observe that for any pair $\{v, w\} \in \binom{X}{2} \setminus \Delta_G(X, S)$, there exists a vertex $u \in S$ such that $\{v, w\} \subseteq N_G(u)$. Consequently, if $\{v, w\} \in E$, then $(u, v, w)$ forms a triangle containing a vertex from $S$; otherwise, $\{v, w\}$ cannot participate in any triangle at all. The procedure PRE-LISTING explicitly handles both cases by considering all triples containing a vertex from $S$. As a result, after its execution, the candidate vertex-pairs for triangles on $X$ can be safely restricted from $\binom{X}{2}$ to $\Delta_G(X, S)$.

Recall that $S$ is the sampled vertex set with $|S| = \Theta(s \log n)$, and $\mathcal{T}_S = \{ T \in \mathcal{T}(G) : T \cap S \neq \emptyset \}$ denote the set of all triangles containing at least one vertex from $S$, with $t_1 := |\mathcal{T}_S|$. Besides, $E_S = \{ \{u, v\} \in E : \exists T \in \mathcal{T}_S, \{u, v\} \subseteq T \}$ denote the set of edges contained in triangles from $\mathcal{T}_S$. Let $t_1'$ denote the number of triangles incident to $E_S$.

**Lemma 3.2** Let $G = (V, E)$ be an $n$-vertex graph. With high probability, PRE-LISTING finds all triangles containing at least one vertex from $S$ in time $\widetilde{O}\left(n s^{1/2} t_1^{1/2} + \sqrt{n t_1 t_1'}\right)$. The initialization for $H_0$ of size $h$ requires $O(sh)$ time. Moreover, after the procedure terminates, every remaining triangle in $G$ is supported entirely on vertex-pairs contained in $\Delta_G(V, S)$.

*Proof.* The initial sampling step requires $O(n)$ time. Next, the procedure applies repeated Grover search over all vertex triples containing at least one vertex from $S$. This corresponds to a search space of size $|S|\binom{n}{2}$, with $t_1$ marked elements. By Lemma C.2, all triangles in $\mathcal{T}_S$ can be listed in $\widetilde{O}\left(\sqrt{|S|\binom{n}{2} \cdot t_1}\right) = \widetilde{O}(n s^{1/2} t_1^{1/2})$ time.

Afterwards, the procedure invokes INCIDENTTRIANGLELISTING to list all triangles incident to edges in $E_S$. By Lemma D.1, this step takes $\widetilde{O}\left(\sqrt{n|E_S| \cdot t_1'}\right)$ time. Note that $|E_S| \leq 3t_1$. Combining the above bounds yields the stated runtime.

We initialize the state $H_0$ together with its associated data structure here, which serves as the starting state for all subsequent quantum walks. $H_0$ can be an arbitrary subset of $V$ with $h = |H_0|$. The data structure only stores pairs $(u, N_G(u) \cap S)$ for $u \in H_0$. Since the set $S$ and $H_0$ is fixed after PRE-LISTING, this information never changes. This requires $\widetilde{O}(|S| \cdot |H_0|) = O(sh)$ time.

Finally, the output consists of all triangles in $\mathcal{T}_S$ together with all triangles incident to $E_S$ (i.e., $\mathcal{T}_S'$). All edges in $\bigcup_{u \in S} \binom{N_G(u)}{2}$ are removed during this process. Therefore, any triangle remaining in $G$ cannot contain a vertex from $S$ and cannot include any vertex-pair with a common neighbor in $S$. By definition, such triangles must be supported entirely on vertex-pairs in $\binom{V}{2} \setminus \bigcup_{u \in S} \binom{N_G(u)}{2} = \Delta_G(V, S)$. $\square$

For any sets $S, X \subseteq V$ and any vertex $w \in V$, define $\Delta_G(X, w, S) \subseteq \Delta_G(X, S)$ by

$$\Delta_G(X, w, S) = \binom{X \cap N_G(w)}{2} \setminus \bigcup_{u \in S} \binom{N_G(u)}{2}$$

$$= \left\{ \{u, v\} \in \Delta_G(X, S) | \{u, w\}, \{v, w\} \in E \right\},$$

That is, $\Delta_G(X, w, S)$ consists of vertex-pairs in $X$ that do not share any common neighbor in $S$, *but* do share $w$ as a common neighbor. We use this refinement to isolate vertex-pairs in $\Delta_G(V, S)$ that are still capable of forming triangles, as only such pairs can participate in triangles with third vertex $w$.

The main purpose of introducing $\Delta_G(X, w, S)$ is to control the number of candidate triples explored during quantum walk search. Naively, searching over all triples containing pairs from $X$ incurs a cost proportional to $n\binom{|X|}{2}$. A well-chosen vertex set $S$ ensures that for every pair surviving PRE-LISTING, the number of possible third vertices is small. This reduces the effective search space by a factor of $s$, which is essential for achieving the desired speedup. We formalize this idea through the notion of *good sets*.

**Definition D.2** (adapted from (Le Gall, 2014)). A set $S \subseteq V$ with size order $s$ is said to be *good* for $G$ if for all $X \subseteq V$,

$$\sum_{w \in V} |\Delta_G(X, w, S)| \leq n|X|^2/s.$$

**Lemma D.3.** *Suppose that $S$ is a set of size $\Theta(s \log n)$ sampled from $V$ uniformly at random with replacement. Then, with high probability, for any $\{u, v\} \in \Delta_G(V, S)$, it holds that*

$$|\{w \in V \mid \{u, w\}, \{v, w\} \in E\}| \leq n/s.$$

*Proof.* For any vertex-pair $\{u, v\} \in \binom{V}{2}$, define $N_{uv} = \{w \in V \mid \{u, w\}, \{v, w\} \in E\}$ as their common neighborhood. Consider any pair $\{u, v\}$ satisfying $|N_{uv}| > n/s$. Take $\Theta(s \log n) = cs \log n$ for some constant $c > 2$. Then, when $S$ is sampled uniformly at random with replacement, the probability that no vertex from $N_{uv}$ is included is

$$\left(1 - \frac{|N_{uv}|}{n}\right)^{\Theta(s \log n)} < \left(1 - \frac{n/s}{n}\right)^{cs \log n} \leq \frac{1}{n^c}.$$

Note that $\{u, v\} \in \Delta_G(V, S)$ where $\Delta_G(V, S) = \binom{V}{2} \setminus \bigcup_{u \in S} \binom{N_G(u)}{2}$ if and only if $S \cap N_{uv} = \emptyset$. Thus, the probability that $\{u, v\} \in \Delta_G(V, S)$ is at most $\frac{1}{n^c}$.

By the union bound, the probability that any pair $\{u, v\}$ with $|N_{uv}| > n/s$ exists in $\Delta_G(V, S)$ is at most $\binom{n}{2} \cdot \frac{1}{n^c}$. Therefore, with probability at least $1 - \frac{1}{n^{c-2}}$, the condition

$$|N_{uv}| = |\{w \in V \mid \{u, w\} \in E \text{ and } \{v, w\} \in E\}| \leq n/s.$$

is satisfied for every pair $\{u, v\} \in \Delta_G(V, S)$. $\qquad\square$

**Lemma D.4.** *With high probability, the set $S$ produced in* PRE-LISTING *is good for $G$.*

*Proof.* For any $X \subseteq V$, we have

$$\sum_{w \in V} |\Delta_G(X, w, S)| = \sum_{w \in V} \left| \left\{ \{u, v\} \in \Delta_G(X, S) \mid \{u, w\}, \{v, w\} \in E \right\} \right|$$

$$= \sum_{\{u, v\} \in \Delta_G(X, S)} |\{w \in V \mid \{u, w\}, \{v, w\} \in E\}|.$$

By Lemma D.3, each pair $\{u, v\} \in \Delta_G(X, S)$ has at most $n/s$ common neighbors. Therefore,

$$\sum_{w \in V} |\Delta_G(X, w, S)| \leq \frac{n}{s} \cdot |\Delta_G(X, S)| \leq \frac{n|X|^2}{s},$$

which completes the proof. $\qquad\square$

### D.1.3. DETAILS AND ANALYSIS OF Q-WALKSEARCH

We now present the details and analysis of the two-level quantum walk subroutine Q-WALKSEARCH (Algorithm 5), based on the quantum triangle finding algorithm of Le Gall (2014).

After the PRE-LISTING step, the task reduces to identifying the remaining triangles. By Lemma 3.2, every such triangle is supported entirely on vertex-pairs contained in $\Delta_G(V, S)$. Consequently, it suffices to search for an edge in $\Delta_G(V, S)$ that participates in at least one remaining triangle. Recall that we denote by $t_2$ the number of such triangles.

At a high level, we employ a two-level quantum walk to locate such an edge. In the *outer* quantum walk, the state space consists of $h$-subsets $H \subseteq V$. For a given $H$, we consider the set $\Delta_G(H, S) \subseteq \binom{H}{2}$ of candidate vertex-pairs that may support a remaining triangle. The set $\Delta_G(H, S)$ can be constructed implicitly from previously queried edges between $H$ and the sampled set $S$, without incurring additional oracle queries.

Given a candidate set $H$, the algorithm then needs to determine whether $\Delta_G(H, S)$ contains an edge that participates in a triangle. To this end, it searches for a vertex $w \in V$ such that $w$ forms a triangle together with some pair in $\Delta_G(H, S)$. For

a fixed choice of $w$, this task is delegated to an *inner* quantum walk, which searches for a smaller subset $H' \subseteq H$ such that $\Delta_G(H', w, S) = \binom{N_G(w)}{2} \cap \Delta_G(H', S)$ contains an actual edge of $G$. If such a pair exists, then it forms a triangle in $G$ together with $w$.

The following lemma summarizes the performance of the inner quantum walk, which is an important technical ingredient in the analysis of Lemma 3.3. Its details and proof is given in Appendix D.1.4.

**Lemma D.5.** *For any fixed $w \in V$, there exists a quantum algorithm with running time*

$$\widetilde{O}\Big(h^{2/3} + \sqrt{|\Delta_G(H, w, S)|}\Big)$$

*that, given a set $H \in \binom{V}{h}$ and the set $\Delta_G(H, S)$, checks with probability at least $1 - 1/\mathrm{poly}(n)$ whether there exists a pair $\{v_1, v_2\} \in \Delta_G(H, S)$ such that $\{v_1, v_2, w\}$ forms a triangle in $G$.*

With a checking procedure for any fixed $H$ and $w$ in place, it remains to search over specific $h$-subset $H \subseteq V$. The following lemma provides the main technical analysis of the *outer* quantum walk.

Lemma 3.3 Given the graph remaining after PRE-LISTING with $n$ vertices and $t_2$ remaining triangles, let $\nu$ be the maximum number of pairwise edge-disjoint triangles in this graph. There exists a quantum walk search algorithm on the Johnson graph $J(V, h)$, starting from an initial state $H_0$ that, with probability at least $1 - 1/\mathrm{poly}(n)$, finds a set $H \subseteq V$ and a vertex $w \in V$ such that $\Delta_G(H, w, S) \cap E \neq \emptyset$. Denote the found set and vertex by $H^*$ and $w^*$. The running time of the algorithm is

$$\widetilde{O}\bigg(\Big(1 + \frac{n}{h\sqrt{\nu}}\Big) \cdot \big(sh^{1/2} + n^{1/2}h^{2/3} + n^{1/2}s^{-1/2}h\big)\bigg),$$

where $s$ is the sample size in PRE-LISTING.

*Proof.* The goal of the outer quantum walk is to find a subset $H \in \binom{V}{h}$ such that $\Delta_G(H, S)$ contains an edge participating in a remaining triangle. Due to the PRE-LISTING step, every remaining triangle contains at least one edge whose endpoints both lie in $\Delta_G(V, S)$; moreover, such an edge appears in $\Delta_G(H, S)$ if and only if $H$ contains both its endpoints. Equivalently, we say a set $H$ is marked if there exists a vertex $w \in V$ such that $\Delta_G(H, w, S) \cap E \neq \emptyset$.

We perform a quantum walk on the Johnson graph $J(V, h)$, whose vertices correspond to $h$-subsets of $V$. The initial state $H_0$ is provided by the main algorithm QW-TRIANGLELISTING. Let $\varepsilon_H$ denote the fraction of marked vertices in $J(V, h)$. The complexity of the walk depends on $1/\sqrt{\varepsilon_H}$, which we now bound using a packing argument.

To avoid the pitfall of treating triangle-supporting edges as independent, we instead work with a maximal set of pairwise edge-disjoint remaining triangles. The following lemma extracts from such a packing a set of certifying edges with low maximum degree.

**Lemma D.6.** *Let $\mathcal{P}$ be a collection of $\nu$ pairwise edge-disjoint triangles in a graph. Then one can choose one edge $e_T$ from each triangle $T \in \mathcal{P}$ such that the graph $F_{\mathcal{P}} := \{e_T : T \in \mathcal{P}\}$ has $|E(F_{\mathcal{P}})| = \nu$ and $d_{\max}(F_{\mathcal{P}}) \leq 3\sqrt{\nu} + 1$, where $d_{\max}(\cdot)$ denotes the maximum degree.*

*Proof.* Call a vertex *heavy* if it belongs to more than $\sqrt{\nu}$ triangles of $\mathcal{P}$. Since the total number of vertex-triangle incidences is $3\nu$, there are at most $3\sqrt{\nu}$ heavy vertices. For each triangle, choose an edge as follows. If the triangle contains at most one heavy vertex, choose an edge joining two non-heavy vertices. If the triangle contains at least two heavy vertices, choose an edge joining two heavy vertices. A non-heavy vertex lies in at most $\sqrt{\nu}$ triangles, so its degree in $F_{\mathcal{P}}$ is at most $\sqrt{\nu}$. A heavy vertex can be incident to a chosen edge only when that edge joins two heavy vertices; because triangles in $\mathcal{P}$ are edge-disjoint, for each other heavy vertex there is at most one such edge. Hence the degree of a heavy vertex in $F_{\mathcal{P}}$ is at most the number of heavy vertices, which is at most $3\sqrt{\nu}$. This establishes the claimed degree bound. $\square$

**Proposition D.7.** *Let $R$ be the residual graph after PRE-LISTING and let $\mathcal{P}$ be a maximum collection of pairwise edge-disjoint triangles in $R$, with $|\mathcal{P}| = \nu \geq 1$. Let $H$ be a uniformly random $h$-subset of $V$. Then*

$$\Pr\big[\Delta_G(H, w, S) \cap E \neq \emptyset \text{ for some } w \in V\big] \geq c \min\Big\{1, \frac{\nu h^2}{n^2}\Big\},$$

*for a universal constant $c > 0$. Consequently,*

$$\frac{1}{\sqrt{\varepsilon_H}} \le C\left(1 + \frac{n}{h\sqrt{\nu}}\right),$$

*where $\varepsilon_H$ is the fraction of marked $h$-subsets in the outer walk, and $C$ is a universal constant.*

*Proof.* Apply Lemma D.6 to $\mathcal{P}$ to obtain a set $F \subseteq E(R)$ of $\nu$ edges such that each edge lies in a distinct triangle of $\mathcal{P}$ and the maximum degree of $F$ is at most $3\sqrt{\nu} + 1$.

Define $Y := |F[H]|$, the number of these edges contained in $H$. If $Y > 0$, then $H$ contains an edge that belongs to some remaining triangle, so $H$ is marked. Thus it suffices to lower bound $\Pr[Y > 0]$.

We have $\mu := \mathbb{E}[Y] = \nu \frac{h(h-1)}{n(n-1)} = \Theta\left(\frac{\nu h^2}{n^2}\right)$. Write $Y = \sum_{e \in F} I_e$ where $I_e = \mathbf{1}_{\{e \subseteq H\}}$. Computing the second moment,

$$\mathbb{E}[Y^2] = \sum_e \mathbb{E}[I_e] + \sum_{e \ne e'} \mathbb{E}[I_e I_{e'}].$$

The contribution of identical edges is $O(\mu)$. Two distinct edges can be adjacent (share a vertex) or disjoint. The number of ordered pairs of adjacent edges in $F$ is $O(\nu \, d_{\max}(F)) = O(\nu^{3/2})$, and for each such pair $\{e, e'\}$ we have $\Pr[e \cup e' \subseteq H] = O(h^3/n^3)$. The number of ordered pairs of disjoint edges is at most $\nu^2$, and for those $\Pr[e \cup e' \subseteq H] = O(h^4/n^4)$. Hence

$$\mathbb{E}[Y^2] \le C\left(\mu + \nu^{3/2}(h^3/n^3) + \nu^2(h^4/n^4)\right).$$

Observe that $\nu^{3/2}(h^3/n^3) = \Theta(\mu^{3/2})$ and $\nu^2(h^4/n^4) = \Theta(\mu^2)$. Therefore $\mathbb{E}[Y^2] \le C'(\mu + \mu^{3/2} + \mu^2) \le C''(\mu + \mu^2)$.

By the Paley–Zygmund inequality,

$$\Pr[Y > 0] \ge \frac{\mathbb{E}[Y]^2}{\mathbb{E}[Y^2]} \ge c\min\{\mu, 1\} \ge c' \min\left\{1, \frac{\nu h^2}{n^2}\right\}.$$

Thus $\varepsilon_H = \Omega(\min\{1, \nu h^2/n^2\})$. The claimed bound on $1/\sqrt{\varepsilon_H}$ follows directly. $\qquad\square$

The data structure associated with a state $H$ in the outer quantum walk is the set $\Delta_G(H, S)$, represented implicitly by storing, for each $u \in H$, the pair $(u, N_G(u) \cap S)$. This information suffices to reconstruct $\Delta_G(H, S)$ without additional oracle queries: a vertex-pair $\{v_1, v_2\} \subseteq H$ belongs to $\Delta_G(H, S)$ if and only if $(N_G(v_1) \cap S) \cap (N_G(v_2) \cap S) = \emptyset$. The data structure is maintained coherently throughout the quantum walk.

The setup cost is $\mathsf{S}_H = 0$, since the initial state $H_0$ and its associated data structure are constructed during the PRE-LISTING step and are provided as input to the walk. The update cost is $\mathsf{U}_H = O(s)$, as transitioning between adjacent states in the Johnson graph requires querying (and unquerying) the neighborhood between the modified vertex and the set $S$. The state size is $r_H = h$. For the checking cost, by Lemma D.5, for any fixed $w \in V$ we can determine whether there exists a pair $\{v_1, v_2\} \in \Delta_G(H, S)$ such that $\{v_1, v_2, w\}$ forms a triangle in time $\widetilde{O}\left(h^{2/3} + \sqrt{|\Delta_G(H, w, S)|}\right)$. Applying variable-cost quantum search (Lemma C.3) over all vertices $w \in V$, the total checking cost is

$$\begin{aligned}
\mathsf{C}_H &= \widetilde{O}\left(\sqrt{\sum_{w \in V}\left(h^{2/3} + \sqrt{|\Delta_G(H, w, S)|}\right)^2}\right) \\
&= \widetilde{O}\left(\sqrt{nh^{4/3} + \sum_{w \in V}|\Delta_G(H, w, S)|}\right) \\
&= \widetilde{O}\left(n^{1/2}h^{2/3} + \sqrt{\sum_{w \in V}|\Delta_G(H, w, S)|}\right) \\
&= \widetilde{O}(n^{1/2}h^{2/3} + n^{1/2}s^{-1/2}h),
\end{aligned}$$

where the last equality follows from the fact that the sampled set $S$ is good (by Lemma D.4). The good-set property holds for all subsets $X \subseteq V$, we have $\sum_{w \in V} |\Delta_G(H, w, S)| \leq \frac{n}{s} h^2$, implying the result.

By the standard analysis of quantum walks on Johnson graphs, the total running time to find a marked state is

$$\mathsf{S_H} + \frac{1}{\sqrt{\varepsilon_H}} \left( \sqrt{r_H}\, \mathsf{U_H} + \mathsf{C_H} \right) = \widetilde{O}\left( \left( 1 + \frac{n}{h\sqrt{\nu}} \right) \cdot \left( sh^{1/2} + n^{1/2} h^{2/3} + n^{1/2} s^{-1/2} h \right) \right),$$

where we used the bound on $\varepsilon_H$ from Proposition D.7. All failure probabilities arising from subroutines such as inner quantum walk are inverse-polynomially small and can be union-bounded, yielding an overall success probability at least $1 - 1/\mathrm{poly}(n)$. $\qquad\square$

After obtaining a marked set $H^*$ and a corresponding vertex $w^*$, a final Grover search(Lemma C.1) over the pairs in $\Delta_G(H^*, S)$ identifies the remaining two vertices of the triangle in time $\widetilde{O}\left( \sqrt{\binom{h}{2}} \right) = \widetilde{O}(h)$. This cost is dominated by the preceding quantum walk term since $\left( 1 + \frac{n}{h\sqrt{\nu}} \right) \cdot (n^{1/2} s^{-1/2} h) \geq h$ for all $s \leq n$, and therefore does not affect the overall asymptotic complexity.

The success probability of the quantum walk can be amplified to $1 - 1/n^4$ by $\Theta(\log n)$ repetitions, which adds only a logarithmic factor to the running time.

**Corollary D.8.** *Let $G$ be the graph remaining after* PRE-LISTING *and let $\nu$ be the maximum number of pairwise edge-disjoint triangles in this graph. If $\nu \geq 1$, then* Q-WALKSEARCH *finds a triangle with high probability in time*

$$\widetilde{O}\left( \left( 1 + \frac{n}{h\sqrt{\nu}} \right) \cdot \left( sh^{1/2} + n^{1/2} h^{2/3} + n^{1/2} s^{-1/2} h \right) \right).$$

### D.1.4. DETAILS AND ANALYSIS OF INNER WALK

We provide the details and analysis for the inner quantum walk appearing in the two-level quantum walk framework here. Given $H$, $S$ and a vertex $w$, the complexity of the inner walk depends on estimating $\Delta_G(H, w, S)$. We begin by recalling a classical estimator from Le Gall (2014), adapted to our setting.

**Lemma D.9** (Lemma 4.2 in (Le Gall, 2014)). *Let $H$ and $S$ be two subsets of $V$, and assume that $\Delta_G(H, S)$ is known. Let $\ell$ be a positive integer. There exists a classical deterministic algorithm $\mathcal{A}$ with running time $\widetilde{O}(\ell \log n)$, which receives as input a binary string $s_r$ of length $\mathrm{poly}(\ell, \log n)$ and a vertex $w \in V$, and outputs a real number $\mathcal{A}(s_r, w)$ satisfying the following condition: with probability at least $1 - \frac{3}{n}$ on the choice of the string $s_r$, the inequalities*

$$\frac{1}{3} \times |\Delta_G(H, w, S)| \leq \mathcal{A}(s_r, w) \leq \frac{3}{2} \times \max \left( \frac{|H| \times (|H| - 1)}{2\ell}, |\Delta_G(H, w, S)| \right) \tag{1}$$

*hold for all vertices $w \in V$.*

We below restate the algorithm SIZEESTIMATING in Le Gall (2014). The algorithm receives as input a vertex $w \in V$ and outputs a real number as the estimation of $\Delta_G(H, w, S)$. Define $\mathcal{A}$ in Lemma D.9 as the deterministic version of SIZEESTIMATING where the bit flips used by SIZEESTIMATING are given to $\mathcal{A}$ as the additional input binary string $s_r$. The correctness of SIZEESTIMATING follows straightforwardly from Le Gall (2014).

To analyze the running time, note that membership in $\Delta_G(H, S)$ is checked in $O(1)$ time since the set is explicitly known. The first counter performs $\lceil 240 \log n \rceil$ iterations, each sampling $\ell$ pairs and checking in $O(\ell)$ time. The second counter performs $\lceil 72\ell \log n \rceil$ iterations of sampling one pair and checking in $O(1)$ time. Thus, $\mathcal{A}'$ runs in total $\widetilde{O}(\ell \log n)$ time.

We now present the analysis of the inner quantum walk omitted before.

Lemma D.5 For any fixed $w \in V$, there exists a quantum algorithm with running time

$$\widetilde{O}\left( h^{2/3} + \sqrt{|\Delta_G(H, w, S)|} \right)$$

that, given a set $H \in \binom{V}{h}$ and the set $\Delta_G(H, S)$, checks with probability at least $1 - 1/\mathrm{poly}(n)$ whether there exists a pair $\{v_1, v_2\} \in \Delta_G(H, S)$ such that $\{v_1, v_2, w\}$ forms a triangle in $G$.

---

**Algorithm 8** SIZEESTIMATING($G, H, \Delta_G(H,S), \ell, w$)

---

1: Initialize counter $c_1 \leftarrow 0$
2: **for** 1 **to** $\lceil 240 \log n \rceil$ **do**
3:     Take $\ell$ elements $\{u_1, v_1\}, \ldots, \{u_\ell, v_\ell\}$ uniformly at random from $\binom{H}{2}$ with replacement
4:     **for** $i = 1$ **to** $\ell$ **do**
5:         **if** $\{u_i, v_i\} \in \Delta_G(H,S)$, $\{u_i, w\} \in E$, and $\{v_i, w\} \in E$ **then**
6:             $c_1 \leftarrow c_1 + 1$
7:         **end if**
8:     **end for**
9: **end for**
10: **if** $c_1 \leq \lceil 240 \log n \rceil / 2$ **then**
11:     **return** $\frac{\binom{|H|}{2}}{\ell}$
12: **else**
13:     Initialize counter $c_2 \leftarrow 0$
14:     **for** 1 **to** $\lceil 72\ell \log n \rceil$ **do**
15:         Take a pair $\{u, v\}$ uniformly at random from $\binom{H}{2}$
16:         **if** $\{u, v\} \in \Delta_G(H,S)$, $\{u, w\} \in E$, and $\{v, w\} \in E$ **then**
17:             $c_2 \leftarrow c_2 + 1$
18:         **end if**
19:     **end for**
20:     **return** $\frac{c_2 \binom{|H|}{2}}{\lceil 72\ell \log n \rceil}$
21: **end if**

---

*Proof.* Fix a vertex $w \in V$. We analyze the algorithm conditioned on the event that the estimator $\mathcal{A}$ from Lemma D.9 satisfies the inequalities in Equation (1), which occurs with probability at least $1 - \frac{3}{n}$. Conditioning on this event, the remainder of the analysis is deterministic.

We set $\ell = \lceil h^{2/3} \rceil$ and invoke $\mathcal{A}$ on input $(\Delta_G(H,S), w)$ to obtain an estimate $\mathcal{A}(s_r, w)$ in $\widetilde{O}(h^{2/3})$ time. By Lemma D.9, this estimate satisfies

$$\mathcal{A}(s_r, w) \leq \frac{3}{2} \times \max\left\{ \frac{h(h-1)}{2\ell}, |\Delta_G(H, w, S)| \right\}.$$

We then perform a quantum walk on the Johnson graph $J(H, \ell)$, whose vertices correspond to $\ell$-subsets $H' \subseteq H$. A subset $H'$ is defined to be marked if it satisfies the following two conditions (as in (Le Gall, 2014)):

(i) $\Delta_G(H', w, S) \cap E \neq \emptyset$, i.e., there exists $\{v_1, v_2\} \in \Delta_G(H', S)$ such that $\{v_1, v_2, w\}$ forms a triangle in $G$;
(ii) $|\Delta_G(H', w, S)| \leq \frac{8(\ell-2)(\ell-3)}{(h-2)(h-3)} \cdot \mathcal{A}(s_r, w) + 16\ell$.

Condition (i) ensures that a marked subset witnesses a triangle involving $w$, while condition (ii) guarantees that the cost of checking markedness remains bounded. Following the analysis of Le Gall (2014), when $\Delta_G(H, w, S) \cap E \neq \emptyset$, the fraction of subsets $H' \subseteq H$ satisfying both conditions is $\varepsilon_{H'} = \Omega(h^{-2/3})$. Intuitively, even in the worst case where $H$ contains a single triangle edge, the probability that both endpoints appear in a random $\ell$-subset is $\Omega((\ell/h)^2) = \Omega(h^{-2/3})$, and the exclusion of subsets violating condition (ii) removes only a lower-order fraction of such successes.

For the quantum walk data structure, we store $\Delta_G(H', w, S)$ implicitly as a collection of pairs $(v, e_v)$ for $v \in H'$, where $e_v = 1$ if $\{v, w\} \in E$ and $e_v = 0$ otherwise. Since $\Delta_G(H, S)$ is given explicitly, this representation suffices to reconstruct $\Delta_G(H', w, S)$ without additional queries. The setup cost is $\mathsf{S}_{H'} = O(|H'|) = O(h^{2/3})$, incurred by querying $\{v, w\} \in E$ for all $v \in H'$. The update cost is $\mathsf{U}_{H'} = O(1)$, and the subset size is $r_{H'} = \ell$. To check whether a subset $H'$ is marked, we proceed as follows. Given $\Delta_G(H', w, S)$, condition (i) can be verified via Grover search in time $\widetilde{O}(\sqrt{|\Delta_G(H', w, S)|})$, while condition (ii) can be checked in $\widetilde{O}(1)$ time. Hence, the checking cost is $\mathsf{C}_{H'} = \widetilde{O}\left(\sqrt{|\Delta_G(H', w, S)|}\right)$.

Applying the standard complexity bound for quantum walks on the Johnson graph (Lemma 2.2), the total running time is

$$\widetilde{O}\left( \ell + \mathsf{S}_{H'} + \frac{1}{\sqrt{\varepsilon_{H'}}}(\sqrt{r_{H'}} \times \mathsf{U}_{H'} + \mathsf{C}_{H'}) \right)$$

$$
= \widetilde{O}\Big(h^{2/3} + h^{2/3} + \frac{1}{h^{-1/3}}\Big(h^{1/3} \times 1 + \sqrt{|\Delta_G(H', w, S)|}\Big)\Big)
$$

$$
= \widetilde{O}\Big(h^{2/3} + h^{1/3} \times \sqrt{h^{-2/3}\mathcal{A}(s_{\mathrm{r}}, w) + h^{2/3}}\Big)
$$

$$
= \widetilde{O}\Big(h^{2/3} + \sqrt{\mathcal{A}(s_{\mathrm{r}}, w) + h^{4/3}}\Big)
$$

$$
= \widetilde{O}\Big(h^{2/3} + \sqrt{|\Delta_G(H, w, S)|}\Big),
$$

where the last equality follows from the the upper bound on $\mathcal{A}(s_{\mathrm{r}}, w)$.

If the estimator $\mathcal{A}$ from Lemma D.9 violates the inequalities in Equation (1), which occurs with probability at most $3/n$, the subsequent quantum walk may incur a large cost. To handle this case, we impose a hard cutoff on the total number of queries and elementary operations allowed. If this cutoff is exceeded, the algorithm halts immediately and outputs that no triangle involving $w$ exists within $\Delta_G(H, S)$. This decision may be incorrect only when the estimator fails, and therefore contributes an additional error probability of at most $3/n$. The quantum walk on the Johnson graph and the Grover search used in the checking procedure each succeed with constant probability. By standard repetition and amplification techniques, we can boost their success probabilities to $1 - O(1/\operatorname{poly}(n))$ at the cost of an additional $\operatorname{poly}\log(n)$ factor in the running time. Combining the above, the overall failure probability is at most $O(1/n)$. □

### D.1.5. OVERALL ANALYSIS OF QW-TRIANGLELISTING

We now remove the temporary assumption that a constant-factor estimate of $t$ is available. Instead of estimating $t$ directly, we try the guesses

$$
\hat{t} = 1, 2, 4, \ldots, 2^{\lceil 3\log_2 n \rceil}.
$$

If an initial amplified triangle-detection procedure finds no triangle in $G$, the algorithm outputs the empty list. Otherwise, since $1 \le t \le n^3$, one of the above guesses satisfies $t \le \hat{t} < 2t$.

For each guess, in increasing order, we run the parameterized version of QW-TRIANGLELISTING with parameters $s, h$ chosen according to this value of $\hat{t}$. Whenever a run outputs a candidate list of triangles, we verify it by running an amplified triangle-detection procedure with an additional QRAM membership test, searching for a triangle not contained in the candidate list. If no such triangle is found, we accept the output; otherwise, we continue to the next guess.

For the guess satisfying $t \le \hat{t} < 2t$, the parameter choices are within constant factors of those obtained using the true value of $t$, so the parameterized analysis applies. Since the guesses increase geometrically and the running-time bound is monotone in $\hat{t}$, all earlier runs contribute only a polylogarithmic overhead. The verification calls also add only polylogarithmic overhead, which is absorbed into the final $\widetilde{O}(\cdot)$ bound.

We now combine the established components to analyze the performance of QW-TRIANGLELISTING in Theorem 3.4.

**Theorem 3.4** Let $G$ be an undirected graph with $n$ vertices and $t$ triangles. There exists a quantum algorithm, QW-TRIANGLELISTING, with high probability, that outputs all triangles $\mathcal{T}(G)$ using

$$
\widetilde{O}\big(n^{5/4}t^{7/12} + n^{7/6}t^{7/9}\big)
$$

elementary operations and queries to the graph oracle $\mathcal{O}_G$.

*Proof of Theorem 3.4.* It suffices to analyze the parameterized version of QW-TRIANGLELISTING under the assumption that it is given $\hat{t} = \Theta(t)$, since the guessing schedule above guarantees such a guess. For readability, the following analysis writes the parameter choices in terms of $t$; replacing $t$ by such an estimate $\hat{t}$ changes the bounds only by constant factors.

The algorithm begins by executing PRE-LISTING, which samples a vertex subset $S$ and enumerates all triangles containing at least one vertex in $S$. It then calls INCIDENTTRIANGLELISTING to find all triangles sharing an edge with the ones already found, and removes those edges. Consequently, every remaining triangle has all three edges contained in $\Delta_G(V, S)$.

Let $\mathcal{T}_S$ be the set of triangles found during PRE-LISTING and $t_1 = |\mathcal{T}_S|$. By Lemma 3.2, this phase costs $\widetilde{O}(ns^{1/2}t_1^{1/2} + \sqrt{nt_1t_1'})$, where $t_1'$ is the number of triangles incident to the edges of $\mathcal{T}_S$. Removing these edges is safe because all triangles containing them have been listed.

Let $t_2$ be the number of triangles remaining after PRE-LISTING. Clearly $t_1, t_1', t_2 \leq t$. The algorithm then repeatedly invokes Q-WALKSEARCH. At the start of the $i$-th quantum-walk iteration, let $\nu_i$ be the maximum number of pairwise edge-disjoint triangles in the current residual graph. Because each successful quantum-walk iteration finds a triangle and then deletes all triangles incident to its three edges, the triangles discovered in different quantum-walk iterations are pairwise edge-disjoint. Therefore, if the quantum-walk phase performs $r$ successful iterations, we have

$$\nu_i \geq r - i + 1 \qquad \text{and} \qquad r \leq \nu_1 \leq t_2.$$

Applying [Corollary D.8](), the cost of the $i$-th iteration is

$$\widetilde{O}\left(\left(1 + \frac{n}{h\sqrt{\nu_i}}\right)\left(sh^{1/2} + n^{1/2}h^{2/3} + n^{1/2}s^{-1/2}h\right)\right).$$

After each successful walk, INCIDENTTRIANGLELISTING enumerates all triangles incident to the newly found triangle, costing $\widetilde{O}(\sqrt{3nt'_{(i)}})$ with $t'_{(i)}$ being the number of such triangles; summing over all iterations and using Cauchy–Schwarz yields a total of $\widetilde{O}(\sqrt{n}\,t_2)$ for this part, which is always dominated by the other terms.

Summing the quantum-walk costs over all $r$ iterations, we obtain

$$\sum_{i=1}^{r} \widetilde{O}\left(\left(1 + \frac{n}{h\sqrt{\nu_i}}\right)\left(sh^{1/2} + n^{1/2}h^{2/3} + n^{1/2}s^{-1/2}h\right)\right)$$

$$= \widetilde{O}\left(\left(r + \frac{n}{h}\sum_{i=1}^{r}\frac{1}{\sqrt{\nu_i}}\right)\left(sh^{1/2} + n^{1/2}h^{2/3} + n^{1/2}s^{-1/2}h\right)\right)$$

$$\leq \widetilde{O}\left(\left(t + \frac{n}{h}\cdot 2\sqrt{t}\right)\left(sh^{1/2} + n^{1/2}h^{2/3} + n^{1/2}s^{-1/2}h\right)\right)$$

$$\leq \widetilde{O}\left(\left(t + \frac{n}{h}t^{2/3}\right)\left(sh^{1/2} + n^{1/2}h^{2/3} + n^{1/2}s^{-1/2}h\right)\right),$$

where the inequality uses $r \leq t$ and $\sum_{i=1}^{r} 1/\sqrt{\nu_i} \leq \sum_{j=1}^{r} 1/\sqrt{j} \leq 2\sqrt{r} \leq 2\sqrt{t}$; for the subsequent optimization it is convenient to further relax the bound using $\sqrt{t} \leq t^{2/3}$ (which holds for all $t \geq 1$).

Adding the preprocessing and incident-listing costs, the total running time is

$$\widetilde{O}\left(ns^{1/2}t^{1/2} + n^{1/2}t + sh + \left(t + \frac{n}{h}t^{2/3}\right)\left(sh^{1/2} + n^{1/2}h^{2/3} + n^{1/2}s^{-1/2}h\right)\right).$$

Each iteration succeeds with high probability; boosting ensures per-iteration failure probability at most $1/n^c (c \geq 4)$. Since the number of iterations is at most $t \leq n^3$, a union bound shows that all quantum-walk iterations succeed with probability $1 - 1/\text{poly}(n)$. Combined with the high-probability guarantees of PRE-LISTING and INCIDENTTRIANGLELISTING, the overall algorithm outputs all triangles with probability $1 - 1/\text{poly}(n)$.

We now optimize the parameters $s$ and $h$.

**Case I: $t \leq n^{3/7}$.** Set $s = n^{1/2}t^{1/6}$ and $h = n^{3/4}t^{1/4}$. In this regime,

$$t + \frac{n}{h}t^{2/3} = \widetilde{O}(n^{1/4}t^{5/12}), \qquad sh^{1/2} + n^{1/2}h^{2/3} + n^{1/2}s^{-1/2}h = \widetilde{O}(nt^{1/6}).$$

The total cost becomes

$$\widetilde{O}\left(n^{5/4}t^{7/12} + n^{1/2}t + n^{5/4}t^{5/12} + n^{1/4}t^{5/12}\cdot nt^{1/6}\right) = \widetilde{O}(n^{5/4}t^{7/12})$$

**Case II: $n^{3/7} < t \leq n^{3/5}$.** Set $s = n^{1/2}t^{1/6}$ and $h = nt^{-1/3}$. Now $t + \frac{n}{h}t^{2/3} = \widetilde{O}(t)$ and $sh^{1/2} + n^{1/2}h^{2/3} + n^{1/2}s^{-1/2}h = \widetilde{O}(n^{7/6}t^{-2/9})$. The running time becomes

$$\widetilde{O}\left(n^{5/4}t^{7/12} + n^{1/2}t + n^{3/2}t^{-1/6} + t\cdot n^{7/6}t^{-2/9}\right) = \widetilde{O}(n^{7/6}t^{7/9}).$$

**Case III:** $t > n^{3/5}$. Set $s = n^{2/3}t^{-1/9}$ and $h = nt^{-1/3}$. Again $t + \frac{n}{h}t^{2/3} = \widetilde{O}(t)$ and $sh^{1/2} + n^{1/2}h^{2/3} + n^{1/2}s^{-1/2}h = \widetilde{O}(n^{7/6}t^{-2/9})$. The running time is

$$\widetilde{O}\left(n^{4/3}t^{4/9} + n^{1/2}t + n^{5/3}t^{-4/9} + t \cdot n^{7/6}t^{-2/9}\right) = \widetilde{O}(n^{7/6}t^{7/9}).$$

In all cases the total cost is bounded by $\widetilde{O}(n^{5/4}t^{7/12} + n^{7/6}t^{7/9})$, which completes the proof. $\qquad\square$

## D.2. Grover Search Approach

**Theorem D.10.** *Let $G$ be an undirected graph with $n$ vertices and $t$ triangles. There exists a quantum algorithm,* GS-TRIANGLELISTING *, with high probability, that outputs all triangles in $\widetilde{O}(n^{3/2}t^{1/2})$ time.*

We describe the algorithm underlying Theorem D.10. To apply quantum search, we define a Boolean function $f \colon V^3 \to \{0,1\}$ by

$$f(u,v,w) = \begin{cases} 1, & \text{if } (u,v),(v,w),(w,u) \in E \\ 0, & \text{otherwise.} \end{cases}$$

Quantum access to $f$ is provided by an oracle $\mathcal{O}_f$, which can be implemented using $O(1)$ queries to the graph oracle $\mathcal{O}_G$. With this oracle, we obtain the following algorithm.

---

**Algorithm 9** GS-TRIANGLELISTING($\mathcal{O}_G$)

---

1: $\mathcal{T}(G) \leftarrow \emptyset$, $V_{\mathrm{vis}} \leftarrow \emptyset$
2: **for** $i = 1$ **to** $n - 2$ **do**
3: $\quad$ Let $\mathcal{R}_i$ denote the set of triples $\{i, v_1, v_2\}$, for all $\{v_1, v_2\} \in \binom{V \setminus V_{\mathrm{vis}}}{2}$
4: $\quad$ $\mathcal{T}_i \leftarrow$ triangles found by repeated Grover search on $\mathcal{R}_i$
5: $\quad$ $\mathcal{T}(G) \leftarrow \mathcal{T}(G) \cup \mathcal{T}_i$
6: $\quad$ $V_{\mathrm{vis}} \leftarrow V_{\mathrm{vis}} \cup \{i\}$
7: **end for**
8: **return** $\mathcal{T}(G)$

---

*Proof.* Recall that $w$ denotes the existence of an edge in an unweighted graph, i.e., $w(u,v) = 1$ if $(u,v) \in E$ and $w(u,v) = 0$ otherwise. For a triple $\{u,v,w\} \in V^3$, we have

$$|\{u,v,w\}\rangle |0\rangle |0\rangle |0\rangle |0\rangle \xmapsto{3\mathcal{O}_G} |\{u,v,w\}\rangle |w(u,v)\rangle |w(v,w)\rangle |w(w,u)\rangle |0\rangle$$

$$\xmapsto{U_{\mathrm{multi}}} |\{u,v,w\}\rangle |w(u,v)\rangle |w(v,w)\rangle |w(w,u)\rangle |f(u,v,w)\rangle,$$

where $U_{\mathrm{multi}}$ satisfies $|a\rangle |b\rangle |c\rangle |0\rangle \xmapsto{U_{\mathrm{multi}}} |a\rangle |b\rangle |c\rangle |abc\rangle$. This procedure uses 3 queries to $\mathcal{O}_G$ and $\widetilde{O}(1)$ additional arithmetic operations, and thus can be implemented efficiently.

GS-TRIANGLELISTING applies Grover search for each vertex $i \in V$. In each iteration, all triangles incident to $i$ are listed in $\mathcal{T}_i$ with probability at least $2/3$, which can be amplified to $1 - O(1/\mathrm{poly})$ by $O(\log n)$ repetitions. Taking a union bound over all iterations, the algorithm outputs the full triangle set $\mathcal{T}(G)$ with high probability. For a fixed vertex $i$, the search space has size $|\mathcal{R}_i|$, and the number of marked elements is $|\mathcal{T}_i|$. By Lemma C.1, the total number of queries to $\mathcal{O}_f$ satisfies

$$\sum_{i=1}^{n-2} \sqrt{|\mathcal{R}_i| \cdot |\mathcal{T}_i|} = \sum_{i=1}^{n-2} \sqrt{\binom{n-i}{2} \cdot |\mathcal{T}_i|} \le \sqrt{\left(\sum_{i=1}^{n-2} \binom{n-i}{2}\right) \cdot \left(\sum_{i=1}^{n-2} |\mathcal{T}_i|\right)} = \sqrt{\binom{n}{3} \cdot t} = O(\sqrt{n^3 t}),$$

which holds due to the Cauchy–Schwarz inequality and the combinatorial identity $\sum_{j=2}^{n-1} \binom{j}{2} = \binom{n}{3}$. Since each query to $\mathcal{O}_f$ can be implemented with $\widetilde{O}(1)$ overhead, the overall time complexity is $\widetilde{O}(n^{3/2}t^{1/2})$. $\qquad\square$

Finally, we discuss the query complexity of our algorithms for triangle listing.

**Query complexity.** Let $Q_{\text{q-list}}$ denote the query complexity of quantum triangle listing, counting only calls to the graph oracles. Since our computational model is based on QRAM, the quantum primitives used in our listing algorithms, including Grover search and quantum walk search, can be implemented with time complexity matching their query complexity up to polylogarithmic factors. Therefore, the stated time bounds also give the corresponding query-complexity upper bounds up to polylogarithmic factors.

For the heavy–light algorithm, there is a slight refinement. The additive $\widetilde{O}(m)$ term in its time complexity comes from explicit neighborhood enumeration. If only graph-oracle queries are counted, this explicit enumeration is unnecessary: the Grover searches can be performed over neighbor indices, and each predicate evaluation queries the required neighbors and edge existence on demand. The remaining global overhead is the $O(n)$ degree queries used to form the heavy–light partition. Hence the heavy–light approach has query complexity $\widetilde{O}\left(n + m^{3/4}t^{1/2}\right)$. Consequently, the combined triangle-listing algorithm admits the query-complexity bound

$$Q_{\text{q-list}} = \widetilde{O}\left(\min\left(n^{5/4}t^{7/12} + n^{7/6}t^{7/9}, n + m^{3/4}t^{1/2}, n^{3/2}t^{1/2}\right)\right).$$

# E. Omitted Algorithms and Analysis in Section 4

*Why the weight of a triangle instance is the product of its edge weights?* To develop an intuition, consider integer-weighted graphs interpreted as unweighted multigraphs where the multiplicity of each edge corresponds to its weight. In this interpretation, the weight of a triangle, which equals the product of its edge weights, is exactly the number of distinct triangle instances in the multigraph. It is natural to extend this definition to real, non-negative weighted graphs.

## E.1. The Strength-Based Classical Algorithm

Kapralov et al. (2022) introduced the notion of motif cut sparsifiers, where a motif is a frequently occurring subgraph, and gave two polynomial-time algorithms for constructing such sparsifiers. In their strength-based approach, each motif is assigned an importance weight, which is defined in terms of its strength. Each iteration then consists of two main steps. First, one estimates the importance weight of each edge as the sum of the importance weights of all motifs containing it. Then, one deterministically retains all edges whose importance weight exceeds a certain threshold, and then applies an independent sampling scheme to the remaining edges, appropriately reweighting the sampled edges.

Here, we specialize their framework to the triangle motif and provide detailed descriptions of the resulting algorithm.

**Lemma E.1** (Strength Estimation, Follows from Theorem 6.1 of (Chekuri & Xu, 2018)). *There exists algorithm* STRENGTH-ESTIMATION *which does the following: it receives as an input a directed weighted graph $G = (V, E, w)$ and the triangle set $\mathcal{T}(G)$ and outputs strength estimations $\kappa'_T$ for each triangle instance $T \in \mathcal{T}(G)$ with the following properties:*

1. *For all $T \in \mathcal{T}(G)$, $\kappa'_T \leq \kappa_T$,*
2. *$\sum_{T \in \mathcal{T}(G)} \frac{w(T)}{\kappa'_T} \leq c(n-1)$, for some constant $c > 0$.*

*The running time of the algorithm is $O(|\mathcal{T}(G)| \log^2 n \log(|\mathcal{T}(G)|))$.*

We now give the formal definitions of the estimated importance weight and the notion of critical edges.

**Definition E.2** (Estimated Importance Weight). Let $G = (V, E, w)$ be an undirected weighted graph and $\mathcal{T}(G)$ denote the set of all triangle instances in $G$. For each triangle instance $T \in \mathcal{T}(G)$, let $\kappa'_T$ be its strength estimate. Then,

- for $T \in \mathcal{T}(G)$, the estimated importance weight is $\widehat{\eta}(T) = w(T)/\kappa'_T$,
- for an edge $e \in E$, the estimated importance weight is $\widehat{\eta}(e) = \displaystyle\sum_{T \in \mathcal{T}(G): e \in E(T)} \widehat{\eta}(T)$.

**Definition E.3** (Critical Edge). Let $G = (V, E, w)$ be an $n$-vertex undirected weighted graph. An edge $e$ of $G$ is called *critical* if its estimated importance weight is at least $\frac{d\varepsilon'^2}{3(\log n + 3)}$.

Here and in the following, we use $d > 0$ to denote a sufficiently small absolute constant, and in the algorithm we choose an absolute constant $c_1$ to govern its success probability. The value of $d$ depends on $c_1$; the precise dependency is given in (Kapralov et al., 2022). The choice of $c_1$ is otherwise arbitrary; for example, we may assume $c_1 = 10$. We also use $\varepsilon'$ to denote an accuracy parameter related to the sparsification error.

We now present the strength-based classical algorithm for constructing an $\varepsilon$-triangle cut sparsifier of graph $G = (V, E, w)$.

---

**Algorithm 10** TRIANGLECUTSPARSIFICATION$(G, \varepsilon)$

---

1: $E_1 \leftarrow E, w_1 \leftarrow w, \varepsilon' \leftarrow \frac{\varepsilon}{15c_1 \log n}$
2: **for** $j = 1$ **to** $\lceil 6c_1 \log n \rceil$ **do**
3:     $G_j \leftarrow (V, E_j, w_j)$
4:     Compute the set $\mathcal{T}(G_j)$ of all triangles in $G_j$
5:     $E_+ \leftarrow \emptyset, E_- \leftarrow \emptyset, w' \leftarrow w_j$
6:     $\{\kappa'_T\}_{T \in \mathcal{T}(G_j)} \leftarrow$ STRENGTHESTIMATION$(G_j, \mathcal{T}(G_j))$
7:     $E_+ \leftarrow E_+ \cup \{e \in E_j : \widehat{\eta}(e) \geq \frac{d\varepsilon'^2}{3(\log n + 3)}\}$
8:     **for** $e \in E_j \setminus E_+$ **do**
9:         **if** a Bernoulli random variable with parameter $p = 2^{-1/6}$ equals 1 **then**
10:             $w'(e) \leftarrow w_j(e)/p$
11:         **else**
12:             $w'(e) \leftarrow 0$
13:             $E_- \leftarrow E_- \cup \{e\}$
14:         **end if**
15:     **end for**
16:     $E_{j+1} \leftarrow E_j \setminus E_-$
17:     $w_{j+1} \leftarrow w'$
18: **end for**
19: **return** $G_{\lceil 6c_1 \log n \rceil + 1}$

---

We first recall the following theorem from Kapralov et al. (2022), which establishes the correctness of the classical algorithm by showing that it indeed produces an $\varepsilon$-triangle cut sparsifier of $G$. Since our quantum algorithm is built on top of this classical algorithm, its correctness analysis follows from that of the classical procedure.

**Theorem E.4** (Lemma 6.3 in (Kapralov et al., 2022)). *Let graph $G = (V, E, w)$ and $\varepsilon \in (0, 1)$ be the input of* TRIANGLE-CUTSPARSIFICATION *and $G'$ be its output. Then, $G'$ is an $\varepsilon$-triangle cut sparsifier of $G$ with probability at least $1 - n^{-c_1 + 5}$ for a sufficiently large $n$.*

In each iteration, the algorithm computes a set $E_+$ containing all critical edges. Furthermore, the size of $E_+$ can be bounded, as shown in the following lemma.

**Lemma E.5** (Corollary 5.1.3 in (Kapralov et al., 2022)). $E_+$ *satisfies* $|E_+| = O\left(\frac{(n-1)\log n}{d\varepsilon'^2}\right)$.

In each iteration, every non-critical edge is independently sampled with probability $p = 2^{-1/6}$. Hence, a fixed non-critical edge survives all $\lceil 6c_1 \log n \rceil$ iterations only if it is sampled in every iteration where it is present, which happens with probability at most $p^{\lceil 6c_1 \log n \rceil} = n^{-\Omega(1)}$. By a union bound over all edges, the probability that at least one such edge is still present at the end of the algorithm is at most $O(1/\operatorname{poly}(n))$. In particular, we obtain the following bound.

**Lemma E.6** (Kapralov et al. (2022), Lemma 6.2). *The output graph of* TRIANGLECUTSPARSIFICATION *contains at most $\widetilde{O}(n/\varepsilon^2)$ edges with probability at least $1 - O(1/\operatorname{poly}(n))$.*

### E.2. Runtime of the Classical Algorithm

Next, we analyze the running time of the classical algorithm. STRENGTHESTIMATION runs in time $O\left(3|\mathcal{T}(G)| \log^2 n \cdot \log(3|\mathcal{T}(G)|)\right)$ (Chekuri & Xu, 2018) per iteration. Let $T_{\text{c-list}}$ denote the time required to classically enumerate all triangles in $G$. Since the algorithm only deletes edges, every triangle in $G$ can be computed only once in time $T_{\text{c-list}}$ at first and then reused throughout the algorithm.

In each iteration, the algorithm computes the values $\widehat{\eta}(e)$ and identifies all critical edges in $O(|E|)$ time, and the sampling step also takes $O(|E|)$ time. Summing over all iterations and absorbing polylogarithmic factors into the $\widetilde{O}(\cdot)$ notation, we obtain the following bound.

**Lemma E.7** (Theorem 4.1 in (Kapralov et al., 2022)). *The running time of* TRIANGLECUTSPARSIFICATION *is* $T_{\text{c-list}} + \widetilde{O}(|E| + |\mathcal{T}(G)|)$, *where* $T_{\text{c-list}}$ *is the time required to classically enumerate all triangles in* $G$, *and* $|\mathcal{T}(G)|$ *is the number of all triangles.*

# F. Applications: Quantum Algorithms for Triangle Clustering

Graph clustering is a fundamental task in machine learning and data mining. While common methods rely on pairwise interactions, many real-world networks often exhibit higher-order structure that is not captured by edges alone. Triangle motifs provide a canonical example of such higher-order connectivity and have been shown to yield substantially distinct community structure in practice and theory (Austin R. Benson & Leskovec, 2016; Tsourakakis et al., 2017; Yin et al., 2018). Triangle clustering extends some well-known clustering algorithms by replacing edge-based conductance with a triangle-based notion that measures how well groups preserve triangular connectivity.

In this section, we show that triangle clustering admits a natural and meaningful quantum speedup by leveraging our quantum algorithm for triangle listing. We present quantum methods that accelerates the main computational bottleneck of motif-based clustering while preserving the structural guarantees as their classical counterparts. We begin by reviewing standard conductance and its triangle-based generalization.

**Definition F.1** (Edge Conductance). Let $G = (V, E, w)$ be an undirected weighted graph. For any nonempty subset $S \subset V$, let $\bar{S} = V \setminus S$. The *edge conductance* of $S$ in $G$ is defined as

$$\phi_G(S) := \frac{\text{cut}_G(S, \bar{S})}{\min\{\text{vol}_G(S), \text{vol}_G(\bar{S})\}},$$

where $\text{cut}_G(S, \bar{S}) = \sum_{u \in S, v \in \bar{S}} w(u, v)$ and $\text{vol}_G(S) = \sum_{u \in S} d_G(u)$, with $d_G(u) = \sum_{v:(u,v) \in E} w(u, v)$.

**Definition F.2** (Triangle Conductance). Let $G = (V, E, w)$ be an undirected weighted graph and $\mathcal{T}(G)$ the set of all triangles in $G$. For any nonempty subset $S \subset V$, let $\bar{S} = V \setminus S$. The *triangle conductance* of $S$ in $G$ is defined as

$$\phi_G^\Delta(S) := \frac{\text{cut}_G^\Delta(S, \bar{S})}{\min\{\text{vol}_G^\Delta(S), \text{vol}_G^\Delta(\bar{S})\}},$$

where

$$\text{cut}_G^\Delta(S, \bar{S}) = \sum_{\substack{T \in \mathcal{T}(G): \\ T \cap S \neq \emptyset \text{ and } T \cap \bar{S} \neq \emptyset}} w(T), \quad \text{vol}_G^\Delta(S) = \sum_{T \in \mathcal{T}(G)} w(T) \cdot |T \cap S|.$$

For an integer $k \geq 2$, a collection $\mathcal{P} = \{S_1, S_2, \ldots, S_k\}$ is a $k$-partition of $V$ if $\bigcup_{i \in [k]} S_i = V$ and $\forall i, j \in [k], S_i \cap S_j = \emptyset$. The $k$-*way conductance* of $G$ is defined as $\phi_k(G) := \min_{\mathcal{P}} \max_{i \in [k]} \phi_G(S_i)$. Similarly, we define the $k$-*way triangle conductance* of $G$ as $\phi_k^\Delta(G) := \min_{\mathcal{P}} \max_{i \in [k]} \phi_G^\Delta(S_i)$.

**Reduction to Edge-Based Clustering.** A key structural property of triangle clustering is that it reduces exactly to standard conductance on an appropriately reweighted graph. The graph defines the weight of each edge as the weight sum of all triangles containing the edge.

**Definition F.3** (Triangle-Weighted Graph). Let $G = (V, E, w)$ and $\mathcal{T}(G)$ be as above. The *triangle-weighted graph* of $G$ is $G_\Delta = (V, E, w')$ where

$$w'(u, v) = \sum_{T \in \mathcal{T}(G):(u,v) \in T} w(T).$$

The following fact (Austin R. Benson & Leskovec, 2016) formalizes the equivalence between triangle conductance and the edge-based one in the corresponding triangle-weighted graph.

**Fact F.4.** *For every integer* $k \geq 2$, *we have* $\phi_k^\Delta(G) = \phi_k(G_\Delta)$.

*Proof.* It suffices to show that for any fixed $k$-partition $\mathcal{P} = \{S_1, S_2, \ldots, S_k\}$ and an index $i \in [k]$, $\phi_G^\Delta(S_i) = \phi_{G_\Delta}(S_i)$. Let $\bar{S}_i = V \setminus S_i$ and $(S_i, \bar{S}_i)$ denote the cut.

Consider a triangle $T \in \mathcal{T}(G)$ with weight $w(T)$. If $T$ crosses the cut $(S_i, \bar{S}_i)$, it contributes exactly $2w(T)$ to $\mathrm{cut}_{G_\Delta}(S_i, \bar{S}_i)$ because there are exactly two of its edges cut by $(S_i, \bar{S}_i)$. Hence, $\mathrm{cut}_{G_\Delta}(S_i, \bar{S}_i) = \sum_{\substack{T \in \mathcal{T}(G): \\ T \cap S_i \neq \emptyset \text{ and } T \cap \bar{S}_i \neq \emptyset}} 2w(T) = 2 \cdot \mathrm{cut}_G^\Delta(S_i, \bar{S}_i)$.

For the volume, note that triangle $T$ contributes $2w(T)$ to $d_{G_\Delta}(v)$ for its vertex $v$ in $S_i$, since $v$ is incident to two edges of $T$, each with weight $w(T)$. Hence, $d_{G_\Delta}(v) = \sum_{T \in \mathcal{T}(G): v \in T} 2w(T)$. Summing over $v \in S_i$,

$$
\begin{aligned}
\mathrm{vol}_{G_\Delta}(S_i) &= \sum_{v \in S_i} \sum_{T \in \mathcal{T}(G): v \in T} 2w(T) \\
&= 2 \sum_{T \in \mathcal{T}(G): v \in T} \sum_{v \in S_i} w(T) \\
&= 2 \sum_{T \in \mathcal{T}(G)} |T \cap S_i| \cdot w(T) \\
&= 2 \cdot \mathrm{vol}_G^\Delta(S_i).
\end{aligned}
$$

Both the cut and volume in $G_\Delta$ are exactly double their triangle-based counterparts in $G$, so the conductance ratios coincide. $\square$

As a consequence, all theoretical guarantees for spectral clustering on $G_\Delta$ transfer directly to triangle clustering on $G$. After listing all triangles $\mathcal{T}(G)$, we can construct the triangle-weighted graph $G_\Delta$ explicitly in QRAM by accumulating, for each triangle $T$, its weight $w(T)$ onto the three incident edges. This requires $\widetilde{O}(|\mathcal{T}(G)|)$ QRAM write operations and is therefore already accounted for in the triangle listing time $T_{\text{q-list}}$. Once $G_\Delta$ is stored, its adjacency-list and vertex-pair queries can be simulated in $\widetilde{O}(1)$ time via QRAM reads. All subsequent clustering algorithms only assume this standard graph-query access to $G_\Delta$.

**Quantum Speedup for Global Triangle Clustering.** Classical motif-based spectral clustering algorithms (Austin R. Benson & Leskovec, 2016; Tsourakakis et al., 2017) first construct the triangle-weighted graph $G_\Delta$ and then apply spectral $k$-means clustering on it. The dominant computational cost is the construction of $G_\Delta$, which is bounded by the time required to enumerate triangles in $G$. Using our quantum triangle listing algorithm, we can build $G_\Delta$ as described above. This satisfies the input requirements of the quantum spectral $k$-means clustering algorithm of Apers & De Wolf (2022). Combining these components yields Corollary 5.1 and Corollary F.5.

**Corollary F.5.** *For any $k \geq 2$, there exists a quantum algorithm that, given query access to $G$, returns a $k$-partition in time* $T_{\text{q-list}} + \widetilde{O}(\sqrt{mn} + n\,\mathrm{poly}(k))$.

When $k > 2$, this method does not have the same Cheeger-like guarantee on quality. We leave it aligned to the classical one, since the output satisfies the same approximation guarantees as classical $k$-way spectral clustering applied to the triangle-weighted graph.

**Quantum Speedup for Local Triangle Clustering.** Local higher-order graph clustering (Yin et al., 2017) studies the problem of finding a low-conductance cluster containing a given seed vertex using motif-based notions of connectivity. In the triangle setting, the classical algorithm first constructs the triangle-weighted graph $G_\Delta$, then computes the approximate personalized PageRank vector for $G_\Delta$, followed by the sweep procedure to output the set with minimal conductance.

**Corollary F.6.** *Given query access to $G$, a seed vertex $s$, a teleportation parameter $\alpha \in (0,1)$, and a tolerance $\varepsilon > 0$, suppose there exists a target cluster $S^\star$ containing $s$ with triangle conductance $\phi_G^\Delta(S^\star)$. Then there exists a quantum algorithm that returns a vertex set $S$ containing $s$ in time $T_{\text{q-list}} + O\left(\frac{1}{\varepsilon(1-\alpha)}\right)$ such that its triangle conductance is at most*

$$
\tilde{O}\left(\min\left\{\sqrt{\phi_G^\Delta(S^\star)}, \frac{\phi_G^\Delta(S^\star)}{\sqrt{1-\alpha}}\right\}\right).
$$

More broadly, many higher-order graph algorithms, such as global spectral clustering, local clustering, and recent motif-based clustering methods for hypergraphs (Italiano et al., 2025), rely on explicit motif enumeration. Our quantum triangle listing algorithm straightforwardly accelerates this shared computational bottleneck.

# G. Lower Bound

In this appendix, we prove an $\Omega(n/\varepsilon^2)$ lower bound on the size of triangle cut sparsifiers via a reduction from the Gap-Hamming-Distance problem, following the classical framework of Andoni et al. (2016).

**Theorem G.1** (Andoni et al. (2016)). *For every integer $n$ and $\varepsilon \in (1/\sqrt{n}, 1)$, there exists an $n$-vertex graph $G$ such that every $\varepsilon$-cut sparsifier of $G$ has $\Omega(n/\varepsilon^2)$ edges.*

The proof of Theorem G.1 is based on a reduction from the one-way distributional communication complexity of the Gap-Hamming-Distance (GHD) problem, which we recall below.

**Theorem G.2** ( Andoni et al. (2016)). *Consider the following distributional communication problem: Alice has as input $n/2$ strings $s_1, \ldots, s_{n/2} \in \{0,1\}^{1/\varepsilon^2}$ of Hamming weight $\frac{1}{2\varepsilon^2}$, and Bob has an index $i \in [n/2]$ together with one string $t \in \{0,1\}^{1/\varepsilon^2}$ of Hamming weight $\frac{1}{2\varepsilon^2}$, drawn as follows:*[3]

- *$i$ is chosen uniformly at random;*
- *$s_i$ and $t$ are chosen uniformly at random but conditioned on their Hamming distance $\Delta(s_i, t)$ being, with equal probability, either $\geq \frac{1}{2\varepsilon^2} + \frac{c}{\varepsilon}$ or $\leq \frac{1}{2\varepsilon^2} - \frac{c}{\varepsilon}$;*
- *the remaining strings $s_{i'}$ for $i' \neq i$ are chosen uniformly at random.*

*Consider a (possibly randomized) one-way protocol, in which Alice sends to Bob an $m$-bit message, and then Bob determines, with success probability at least $2/3$, whether $\Delta(s_i, t)$ is $\geq \frac{1}{2\varepsilon^2} + \frac{c}{\varepsilon}$ or $\leq \frac{1}{2\varepsilon^2} - \frac{c}{\varepsilon}$. Then Alice's message size is $m \geq \Omega(n/\varepsilon^2)$ bits.*

Using Theorem G.2, Andoni et al. (2016) construct a family of bipartite graphs such that any cut sparsifier with $m$ edges induces a one-way protocol with nearly $m$ bits of communication. This yields Theorem G.1.

We now extend this lower bound to triangle-weighted graphs.

**Lemma G.3.** *For every integer $n$ and $\varepsilon \in (1/\sqrt{n}, 1)$, there exists an $n$-vertex triangle-weighted graph $G_\Delta$ such that every $\varepsilon$-cut sparsifier of $G_\Delta$ has $\Omega(n/\varepsilon^2)$ edges.*

*Proof.* We reduce from the one-way distributional Gap-Hamming-Distance problem in Theorem G.2. Fix $n$ and $\varepsilon$. Given Alice's input strings $s_1, \ldots, s_{n/2} \in \{0,1\}^{1/\varepsilon^2}$, we first recall the bipartite graph construction used in Andoni et al. (2016). The graph consists of $\varepsilon^2 n/2$ disjoint components. Each component is a bipartite graph with left side $L$ and right side $R$, where $|L| = |R| = 1/\varepsilon^2$. Each vertex $u \in L$ corresponds to one string $s_u$, and is connected to exactly those vertices in $R$ indicated by the 1-entries of $s_u$. All edges have unit weight. As shown in Andoni et al. (2016), any $\varepsilon$-cut sparsifier of this graph with $m$ edges yields a one-way protocol for Theorem G.2 using nearly $m$ bits.

We now transform each bipartite component into a tripartite graph still based on the strings $s_1, \ldots, s_{n/2}$. For each component, we construct a graph $G_j$ with vertex partition

$$L_1(G_j) \ \cup \ L_2(G_j) \ \cup \ R(G_j),$$

where

$$|L_1(G_j)| = |L_2(G_j)| = |R(G_j)| = 1/\varepsilon^2.$$

Vertices in $L_1(G_j)$ correspond one-to-one with the vertices of $L$ in the original bipartite construction. Edges are added as follows:

- For each $u \in L_1(G_j)$ and $v \in R(G_j)$, include edge $(u, v)$ of weight 1 if and only if the corresponding bit in $s_u$ is 1.
- For each $u \in L_1(G_j)$, let $u'$ denote its corresponding vertex in $L_2(G_j)$, and add the edge $(u, u')$ of weight 1.
- For each edge $(u, v)$ between $L_1(G_j)$ and $R(G_j)$, add the edge $(u', v)$ between $L_2(G_j)$ and $R(G_j)$ with weight 1.

In other words, we derive the graph by copying the left part of the original graph as $L_2(G_j)$ together with its edges connected with $R(G_j)$. Then, add an edge from each vertex in $L_1(G_j)$ to the corresponding vertex in $L_2(G_j)$. Let $G$ denote the disjoint union of all such components $G_j$. The total number of vertices is $\Theta(n)$.

---

[3] Alice's input and Bob's input are *not* independent, but the marginal distribution of each one is uniform over its domain, namely, $\{0,1\}^{(n/2) \times (1/\varepsilon^2)}$ and $[n] \times \{0,1\}^{1/\varepsilon^2}$, respectively.

Let $G_\Delta$ be the triangle-weighted graph of $G$, where each edge weight equals the number of triangles in $G$ containing that edge. By construction, every triangle in $G$ consists of exactly one vertex from each of $L_1(G_j)$, $L_2(G_j)$, and $R(G_j)$. For each $u \in L_1(G_j)$, the edge $(u, u')$ is contained in exactly $\deg(u)$ triangles, where $\deg(u)$ is the degree of $u$ into $R(G_j)$, which equals the Hamming weight of $s_u$. Thus, in $G_\Delta$, the edge $(u, u')$ has weight $1/(2\varepsilon^2)$. All other edges inherit weights that exactly match the original bipartite construction.

Suppose there exists an $\varepsilon$-cut sparsifier $G'_\Delta$ of $G_\Delta$ with $m = o(n/\varepsilon^2)$ edges. We show that it induces an $\varepsilon$-cut sparsifier of the original bipartite graph $H$ with at most $m$ edges, contradicting Theorem G.1.

Construct a projected weighted graph $\Pi(G'_\Delta)$ on the vertex set of $H$ as follows. Every edge of $G'_\Delta$ between $L_1(G_j)$ and $R(G_j)$ is mapped to the corresponding edge of $H$ with the same weight. Every edge of $G'_\Delta$ between $L_2(G_j)$ and $R(G_j)$ is also mapped to the corresponding edge of $H$ by identifying each vertex $u' \in L_2(G_j)$ with its copy $u \in L_1(G_j)$. Edges between $L_1(G_j)$ and $L_2(G_j)$ are discarded, and parallel projected edges are merged by summing their weights. Thus $\Pi(G'_\Delta)$ has at most $m$ edges.

It remains to verify that $\Pi(G'_\Delta)$ preserves all cuts of $H$. For any cut $(S, \bar{S})$ of $H$, consider the synchronized cut $(S_{\text{sync}}, \overline{S_{\text{sync}}})$ in $G_\Delta$ that places each pair $u \in L_1(G_j)$ and $u' \in L_2(G_j)$ on the same side, and places every $v \in R(G_j)$ according to its side in $(S, \bar{S})$. Under this synchronized cut, the matching edges between $L_1(G_j)$ and $L_2(G_j)$ do not cross, while the two edge types $L_1(G_j)$–$R(G_j)$ and $L_2(G_j)$–$R(G_j)$ contribute two identical copies of the original bipartite cut. Hence $\text{cut}_{G_\Delta}(S_{\text{sync}}, \overline{S_{\text{sync}}}) = 2\,\text{cut}_H(S, \bar{S})$. By the definition of the projection, the cut value of $\Pi(G'_\Delta)$ on $(S, \bar{S})$ is exactly the cut value of $G'_\Delta$ on the synchronized cut.

Since $G'_\Delta$ is an $\varepsilon$-cut sparsifier of $G_\Delta$, the projected graph $\Pi(G'_\Delta)$ preserves every cut of $H$ within a factor of $(1 \pm \varepsilon)$ after the common factor 2. Therefore, scaling all edge weights of $\Pi(G'_\Delta)$ by $1/2$ gives an $\varepsilon$-cut sparsifier of $H$ with at most $m = o(n/\varepsilon^2)$ edges. This contradicts Theorem G.1. Therefore, every $\varepsilon$-cut sparsifier of $G_\Delta$ has $\Omega(n/\varepsilon^2)$ edges. $\qquad\square$

Finally, we relate cut sparsifiers of triangle-weighted graphs to triangle cut sparsifiers of the underlying graph.

**Theorem 5.2**  For every integer $n$ and $\varepsilon \in (1/\sqrt{n}, 1)$, there exists an $n$-vertex graph $G$ such that every $\varepsilon$-triangle cut sparsifier of $G$ has $\Omega(n/\varepsilon^2)$ edges.

*Proof.*  Let $G$ be the graph constructed in the proof of Lemma G.3, and let $G_\Delta$ be its corresponding triangle-weighted graph. Suppose, for contradiction, that there exists an $\varepsilon$-triangle cut sparsifier $G'$ of $G$ with $o(n/\varepsilon^2)$ edges. Let $(G')_\Delta$ be the triangle-weighted graph induced by $G'$.

By Fact F.4, for every cut $(S, \bar{S})$,
$$\text{cut}_{G_\Delta}(S, \bar{S}) = 2 \cdot \text{cut}_G^\Delta(S, \bar{S})$$

and
$$\text{cut}_{(G')_\Delta}(S, \bar{S}) = 2 \cdot \text{cut}_{G'}^\Delta(S, \bar{S}).$$

Since $G'$ is an $\varepsilon$-triangle cut sparsifier of $G$, we have
$$\text{cut}_{G'}^\Delta(S, \bar{S}) = (1 \pm \varepsilon)\,\text{cut}_G^\Delta(S, \bar{S})$$

for every cut $(S, \bar{S})$. Therefore,
$$\text{cut}_{(G')_\Delta}(S, \bar{S}) = (1 \pm \varepsilon)\,\text{cut}_{G_\Delta}(S, \bar{S}).$$

Hence $(G')_\Delta$ is an $\varepsilon$-cut sparsifier of $G_\Delta$. Moreover, every edge of $(G')_\Delta$ is an edge of $G'$, so
$$|E((G')_\Delta)| \le |E(G')| = o(n/\varepsilon^2).$$

This contradicts Lemma G.3. $\qquad\square$

