# OpenReview forum: "Quantum Algorithms for Triangle Cut Sparsification"
_ICML.cc/2026/Conference — ICML 2026 regular_

### Official Review · Reviewer_kuVA · 2026-02-21

**Soundness:** 4
**Presentation:** 3
**Significance:** 3
**Originality:** 3
**Overall Recommendation:** 5
**Confidence:** 4

**Summary:**

The paper proposes quantum algorithms for triangle cut sparsification: building a small reweighted subgraph that approximately preserves, for every cut, the total weight of triangles crossing that cut. It contributes (1) a faster quantum triangle-listing routine in several parameter regimes, and (2) an end-to-end quantum method that constructs an \varepsilon-triangle cut sparsifier with \tilde O(n/\varepsilon^2) edges and runtime \tilde O(T_{q\text{-list}}+\sqrt{mn}/\varepsilon). It also proves an \Omega(n/\varepsilon^2) lower bound on sparsifier size, showing the output sparsity is essentially optimal up to log factors.

**Compliance With Llm Reviewing Policy:**

Affirmed.

**Final Justification:**

I recommend acceptance. The paper is technically sound, clearly presented, and addresses an interesting problem; the rebuttal adequately addressed my main concerns and reinforced my positive assessment.

**Key Questions For Authors:**

What is the essential role of quantum computation here—compared to the best classical graph algorithms, in which regimes (and under which data-access assumptions) does the quantum approach yield a clear advantage, and is the quantum component necessary rather than a wrapper around existing primitives?

**Limitations:**

(1) Accessibility: the query/oracle model and basic quantum notation are not sufficiently explained for non-quantum readers; a brief primer where the notation first appears would help. (2) Novelty clarity: the paper builds on many prior techniques, so it should more explicitly separate the genuinely new ideas/results from reused components. (3) Lower-bound discussion: it would strengthen the paper to briefly discuss quantum query lower-bound tools beyond reductions (e.g., adversary or polynomial methods), and whether/how such techniques might apply to triangle listing/sparsification.

**Strengths And Weaknesses:**

Overall, the paper is technically solid and clearly structured: it formalizes triangle cut sparsification, proposes a quantum triangle-listing primitive with improved complexity in several regimes, and builds an end-to-end quantum algorithm that outputs an \varepsilon-triangle cut sparsifier with near-optimal size (up to log factors), complemented by a matching-style sparsity lower bound. The problem itself is reasonable and relevant to higher-order (triangle/motif) structure in graphs, and the pipeline from triangle access to sparsification is coherent.

That said, I think the paper should better justify why quantum is necessary and what role it plays beyond providing a faster subroutine: compared to the strongest classical graph algorithms, in which regimes does the quantum approach yield clear advantages, and could the quantum component be “removed” without changing the main message? For ICML readers without a quantum background, a short clarification of the query/oracle model and basic notation (e.g., around where quantum symbols first appear) would help. Finally, since the approach builds on a number of existing techniques, it would help to more explicitly clarify what the main new technical ideas are. I am also curious about the landscape of quantum query lower-bound tools for problems like triangle listing/sparsification—beyond reduction-based arguments; adding a brief discussion of what techniques might apply here and what the main obstacles are would strengthen the paper.

---

> ### Author Rebuttal · Authors · 2026-03-31
>
> We sincerely thank the reviewer for their thorough evaluation and insightful suggestions. We address the main points below.
>
> **On the essential role of quantum computation and regimes of advantage.** The quantum component is not a wrapper; it directly accelerates the two bottlenecks in the classical strength-based sparsifier framework: triangle listing and post-sampling. Our results are obtained in the standard quantum query model with QRAM access, where adjacency queries can be performed in superposition. This data-access model is common in quantum graph algorithms and is crucial for enabling quantum primitives to achieve speedups. For example, Grover-type search provides a quadratic speedup over search spaces, reducing $n^3$ to $n^{3/2}$ in the relevant terms.
>
> Concretely, our improvements hold across several regimes:
>
> - Moderate triangle regime. When $t \leq n^{15/14}$, our bound matches or improves the conjectured classical bound $\tilde{O}(n^2 + nt^{2/3})$ (under $\omega = 2$). Quantum algorithms for triangle finding have led to a long line of work developing key techniques (e.g., learning graphs, nested quantum walks). Our work exploits triangle structure together with quantum walk techniques and carefully adapts Le Gall's framework from detection to listing, which is why this method is particularly efficient when $t$ is moderate.
>
> - Sparse graphs. For sparse graphs, the term $m + m^{3/4}t^{1/2}$ dominates and is no larger than the corresponding classical bound. This leverages that low-degree vertices restrict the search space of potential triangles.
>
> - Dense triangle regime. For large $t$, the Grover-based term $\tilde{O}(n^{3/2}t^{1/2})$ improves classical bounds in certain regimes of $\omega$. When $t=\Theta(n^3)$, the output size itself dominates, limiting further speedups.
>
> Together with our speedup for post-sampling, the full sparsification pipeline (Theorem 1.2) inherits these gains. Removing quantum components would revert the runtime to $\tilde{O}(T_{\text{c--list}} + m)$, eliminating the polynomial improvements.
>
> **On technical novelty.** While our approach builds on prior work (e.g., Kapralov et al., 2022; Apers & de Wolf, 2022), the contributions are nontrivial integrations and extensions:
> (i) a new quantum triangle-listing algorithm combining heavy-light decomposition with a generalized two-level quantum walk, extending detection to listing nontrivially (e.g., removing incident edges without omission, repeatedly finding all triangles without duplication;
> (ii) an end-to-end integration with speedup for post-sampling into triangle cut sparsification; and
> (iii) a matching $\Omega(n/\varepsilon^2)$ lower bound on sparsifier size.
>
> **On accessibility.** We thank the reviewer for the helpful advice to improve accessibility. While some background is currently in Appendix C, we will add a short primer on the query/oracle model and clarify notation at first use in the main text.
>
> **On lower-bound techniques.** We also thank the reviewer for this constructive suggestion. Currently, we prove a matching $\Omega(n/\varepsilon^2)$ lower bound on sparsifier size via reduction.
> For related problems:
> - An $\Omega(n)$ quantum query lower bound for triangle finding is known via reduction to the OR problem, which is not tight compared to the $(n^{5/4})$ upper bound. There may be a chance to directly use adversary or polynomial methods to derive better lower bound.
> - For triangle listing, no nontrivial quantum lower bound is known. Existing adversary or polynomial methods are primarily designed for Boolean decision problems. In contrast, triangle listing is a global structure recovery problem with non-constant output size ($\Theta(t)$), which unmatches the problem structure and limits the applicability of such methods.
>
> For sparsification:
> - For edge cut sparsifiers, a quantum query lower bound of $\Omega(\sqrt{mn}/\varepsilon)$ is known (Kapralov et al., 2022). It is proved by reducing sparsification to a structured search problem where a large fraction of edges from some unsparsifiable graphs (hard instance) must be found among the whole graph. This search task is then formulated as a composition of finding marked entries in a Boolean matrix with the OR function. The final bound follows by applying a composition theorem for the general adversary method, combining lower bounds of the subproblems.
> - For hypergraph sparsification, a lower bound of $\Omega(r\sqrt{mn}/\varepsilon)$ ((where $r$ is the rank)) has been conjectured (Liu et al., 2025).
>
> Extending these techniques to triangle cut sparsification remains nontrivial. Based on the framework for the edge case and our size lower bound $\Omega(n/\varepsilon^2)$ (an unsparsifiable graph instance), it is natural to conjecture a lower bound of $\Omega(\sqrt{mn}/\varepsilon)$, which we leave as an open question. We will include this discussion in the revision.
>
> We thank the reviewer again for the constructive feedback and discussion about lower bounds.

---

> > ### Author Rebuttal · Reviewer_kuVA · 2026-04-01
> >
> > Thanks for the response. I raised my score from 4 to 5.

---

### Official Review · Reviewer_UMJX · 2026-03-11

**Soundness:** 3
**Presentation:** 2
**Significance:** 3
**Originality:** 3
**Overall Recommendation:** 4
**Confidence:** 3

**Summary:**

This paper studies quantum algorithms to sparsify the graph while preserving the triangle weights across the cut. Classical algorithm uses $\widetilde O(T\_{\rm c-list}+n^\omega+m)$ time, and the proposed algorithm runs in $\widetilde O(T\_{\rm q-list}+\sqrt{mn}/\epsilon)$ time, where $T\_{\rm q-list}$ is smaller than $T\_{\rm c-list}$ in most parameter regimes. Prior works focus on quantum algorithms to detect triangles, but triangle cut sparisfier is much more involved, as it requires to list triangles. To do so, the algorithm first applies repeated Grover search on the heavy-light vertex partition, then extending the quantum triangle detection to listing via a quantum walk: it first randomly samples a subset of vertices and uses repeated Grover to find all triangles with at least one vertex in the subset, and remove all such triangles with other triangles that share an edge in this set. This is called the pre-listing phase. After this phase, one applies a two-level quantum walk over the Johnson graph to list remaining triangles. To compute the triangle cut sparsifier, one “quantumizes” the motif cut sparsifier algorithm by finding the critical edges via Grover, and sampling the remaining edges via Grover. The developed algorithm could also be used for triangle clustering, and they also prove a lower bound on the linear size of the sparsifier.

**Compliance With Llm Reviewing Policy:**

Affirmed.

**Final Justification:**

I'm satisfied with the response, and I'd like to maintain my score.

**Key Questions For Authors:**

I have two questions.

1. Do you think such a speedup could be further obtained for general motif sparsifiers, or the techniques introduced are specialized for triangles?

2. Do you think a similar speedup could be obtained for 4-cycle listings?

**Limitations:**

Authors are encouraged to provide more detailed discussions on the QRAM model.

**Strengths And Weaknesses:**

### **Strengths**

1. Computing a motif, more specifically triangle cut sparsifier is an interesting problem, and using Grover-based approach to achieve a speedup (most evidently $n^\omega+m$ to $\sqrt{mn}/\epsilon$) is quite important.

2. The algorithm is not simply a translation from classical to quantum, it is a careful examination and combination of several quantum components with different techniques. The components such as the quantum triangle listing, and the algorithm for finding critical edges might be of independent interest.

### **Weaknesses**

1. The model is the QRAM model, which is one of the “stronger” quantum computing models. It is known that to simulate $k$ bits access to QRAM by increasing the number of gates and qubits in the regular circuit model, hence increasing the time complexity by a factor of $k$. Therefore, it is important to measure the query complexity of the algorithm. I strongly recommend authors to discuss the limitations of QRAM and add the query complexity of their algorithm.

2. I would only classify the problem studied in this paper as weakly related to ICML, as triangle cut sparisfier is really a problem most interesting to TCS community, though one could argue that it has some relevance to sparsify and cluster the graph via community structures. I feel theory conferences are more suitable for this work.

---

> ### Author Rebuttal · Authors · 2026-03-31
>
> We sincerely thank the reviewer for the professional review and for recognizing the nontrivial integration of multiple techniques.
>
> **On the QRAM model and query complexity.** We would like to clarify that assuming QRAM access is standard in quantum algorithms for graph problems (e.g., Apers & De Wolf, 2022 [1] and subsequent works), where it plays a role analogous to the classical RAM model by enabling efficient access to input data. Our results follow this widely adopted framework for analyzing asymptotic quantum speedups.
>
> In this setting, query complexity measures the number of accesses to the input, while time complexity additionally accounts for elementary gates and QRAM operations. The role of QRAM is to enable efficient implementation of queries and primitives (e.g., quantum walks), so that query complexity translates to time complexity up to polylogarithmic factors. Accordingly, our algorithms admit a natural query complexity characterization, and thus the query complexity matches the stated time complexity.
>
> We agree that QRAM is a stronger model. We will include a more explicit discussion of this assumption, as well as the corresponding limitations and query complexity, in the revision.
>
> **On ICML relevance.** The problem we study, triangle (motif) cut sparsification, is directly motivated by higher-order graph learning tasks such as triangle-based clustering and motif-aware community detection. These rely on efficient triangle listing and sparsification, and our work provides provable quantum speedups for these core primitives, improving the scalability of downstream ML pipelines. Methodologically, quantum algorithms naturally accelerate large combinatorial search problems (e.g., via Grover search and quantum walks). Our contribution is thus a systematic acceleration of key building blocks in graph-based learning, rather than a superficial use of quantum tools.
>
> Finally,  our work aligns with a growing line of research at the intersection of quantum algorithms and machine learning. Recent examples include quantum speedups for hypergraph sparsification at ICML 2025 (Liu et al., 2025 [2]) and related works (Liu et al., 2024 [3]; Peng & Xu, 2025 [4]), reflecting increasing interest in "quantum for AI". Our work extends this direction to triangle-based structures.
>
> **On extensions to general motifs.** We agree that extending our techniques beyond triangles is an important future direction. Our framework can be partially generalized: in particular, the sampling and reweighting stage after identifying “critical” edges is not specialized for some specific motif. However, the main bottleneck lies in the motif listing primitive.
>
> Our current approach exploits structural properties specific to triangles—for example, that a triangle is determined by a vertex together with a pair of its neighbors—which enables efficient use of  Grover search and heavy-light decomposition. For more complex motifs, such as 4-cycles or higher-order patterns, analogous structural primitives are less direct, and it is not clear how to obtain similar speedups in general. We will add a brief discussion of these challenges in the revision.
>
> **On 4-cycle listing.** Existing classical algorithms for listing 4-cycles (Abboud et al., 2023 [5]) rely on two main approaches. One approach enumerates all length-2 paths and observes that two such paths with the same endpoints form a 4-cycle, leading to a runtime of $O(n^2 + t)$, where $t$ is the number of 4-cycles. In this approach, the main cost comes from explicitly constructing and aggregating these 2-paths (whose total number can be as large as $O(n^2)$), and the dependence on $t$ reflects the inherent output size, making it difficult to obtain significant quantum speedups.
>
> Another approach is based on heavy-light vertex partitioning, achieving a runtime of $O(m^{4/3} + t)$. Since our algorithm also builds on heavy-light techniques, some components may be relevant in this regime. However, extending our approach to 4-cycles is nontrivial, as the structural primitives used for triangles do not directly generalize.
> Overall, while there may be partial connections, achieving comparable quantum speedups for 4-cycle listing remains an interesting open direction.
>
> We thank the reviewer again for the insightful feedback and suggested future extensions.
>
> [1] Apers, S., & De Wolf, R. *Quantum speedup for graph sparsification, cut approximation, and Laplacian solving.* SICOMP 2022.
> [2] Liu, C., Gao, M., Ji, Z., & Ying, M. *Quantum speedup for hypergraph sparsification.* ICML 2025.
> [3] Liu, C., Guan, C., He, J., & Lui, J. C. *Quantum algorithms for non-smooth non-convex optimization.* NeurIPS 2024.
> [4] Peng, P., & Xu, H. *Differentially private synthetic graphs preserving triangle-motif cuts.* COLT 2025.
> [5] Abboud, A., Khoury, S., Leibowitz, O., & Safier, R. *Listing 4-Cycles.* FSTTCS 2023.

---

> > ### Author Rebuttal · Reviewer_UMJX · 2026-04-01
> >
> > I like to maintain my positive score.

---

### Official Review · Reviewer_nM8w · 2026-03-12

**Soundness:** 3
**Presentation:** 3
**Significance:** 3
**Originality:** 3
**Overall Recommendation:** 5
**Confidence:** 1

**Summary:**

This paper explores quantum algorithms for triangle cut sparsification, a technique used to reduce a graph's size while approximately preserving the number of triangles across every cut. The authors present a new quantum algorithm for triangle listing that achieves a polynomial speedup over the best known classical bounds by utilizing methods like heavy-light vertex partitioning and Grover search. Leveraging this listing primitive, they develop a quantum algorithm that constructs triangle cut sparsifiers in $O(T_{q-list} + \sqrt{mn}/\epsilon)$ time. They also demonstrate the practical utility of these methods by applying them to triangle-based clustering algorithms, which inherit the same quantum speedups. Finally, the work establishes a lower bound of $\Omega(n/\epsilon^2)$ for the size of triangle cut sparsifiers, proving that their algorithm's output is nearly optimal

**Compliance With Llm Reviewing Policy:**

Affirmed.

**Key Questions For Authors:**

Practical Threshold for Quantum Advantage: In your complexity analysis, you demonstrate a polynomial speedup over classical bounds ($T_{q-list}$). However, quantum algorithms often carry significant overhead due to error correction and state preparation. Could you provide an estimate or discussion on the graph size ($n$) or triangle density ($t$) at which your quantum algorithm would realistically outperform a state-of-the-art classical implementation on modern hardware?

Robustness to Real-World Graph Noise: Your algorithms provide high-probability guarantees for accurate sparsification. In many machine learning applications, the input graph data is "noisy" (missing edges or containing spurious triangles). How sensitive is the triangle cut sparsifier to small perturbations in the input graph compared to traditional edge-based sparsifiers?

Robustness to Real-World Graph Noise: Your algorithms provide high-probability guarantees for accurate sparsification. In many machine learning applications, the input graph data is "noisy" (missing edges or containing spurious triangles). How sensitive is the triangle cut sparsifier to small perturbations in the input graph compared to traditional edge-based sparsifiers?

**Limitations:**

Yes

**Strengths And Weaknesses:**

Strengths

Algorithmic Innovation: The paper successfully adapts classical graph techniques (like heavy-light vertex partitioning) into the quantum realm, creating a superior triangle listing primitive that outperforms current classical bounds.

Theoretical Optimality: The authors don’t just provide an algorithm; they prove a matching lower bound of $\Omega(n/\epsilon^2)$. This demonstrates that their triangle cut sparsifier is nearly as small as theoretically possible.

Broad Utility: The significance is high because the results directly accelerate practical downstream machine learning tasks, such as triangle-based spectral clustering and community detection.

Rigorous Soundness: The complexity analysis is thorough, using high-probability guarantees ($1 - poly(1/n)$) and clear proofs for each stage of the quantum sampling process.

Weaknesses

Parameter Sensitivity: While the algorithm improves on classical bounds over a "broad range" of parameters, its advantage is less pronounced in extremely dense or extremely sparse graphs where specialized classical heuristics might still be competitive.

---

> ### Author Rebuttal · Authors · 2026-03-31
>
> We sincerely thank the reviewer for the positive evaluation and for highlighting the algorithmic novelty, theoretical optimality, and connections to downstream machine learning tasks.
>
> **On the practical threshold for quantum advantage.** Our results are developed in the standard QRAM model, which focuses on asymptotic complexity and abstracts away hardware-level overheads. Therefore, our guarantees should be interpreted as asymptotic speedups rather than immediate practical improvements on near-term devices.
>
> That said, our algorithms achieve polynomial improvements in the dominant bottleneck, triangle listing, which is computationally expensive even classically. For large-scale graphs (e.g., with millions or billions of edges), such improvements suggest that, once sufficiently large-scale fault-tolerant quantum hardware becomes available, the crossover point could occur at practically relevant input sizes. We will clarify this interpretation in the revision.
>
> **On robustness to noisy graphs.**  We thank the reviewer for this good and insightful question. We interpret “robustness” as follows: if the input graph undergoes a small perturbation, then the resulting sparsifier should not change significantly compared to that of the original graph. The answer depends on the nature of the perturbation.
>
> For *random or unstructured noise* (e.g., randomly missing edges), the effect is typically mild. Such perturbations resemble the post edge-sampling step already performed in the sparsification procedure, and therefore have limited impact on the resulting sparsifier. In this sense, the construction is relatively robust to small random perturbations.
>
> In contrast, for *adversarial perturbations*, the problem can be inherently sensitive. For example, consider a graph where a single edge participates in all the triangles in the graph. Removing this “critical” edge can destroy all the triangles and significantly change triangle-based cut values. Consequently, the resulting sparsifier may differ substantially (e.g., losing a large number of edges that previously carried large weight).
>
> This phenomenon is not specific to triangle cut sparsification: classical edge cut sparsifiers exhibit the same behavior—if a perturbation significantly changes the underlying cut structure, then any sparsifier must reflect this change.
>
> Overall, a deeper understanding of sensitivity under different noise models is an interesting direction, and we need deeper insight to have a thorough understanding.
>
> **On parameter sensitivity.** We agree that, as with most triangle listing algorithms (both classical and quantum), different parameter regimes favor different techniques. Our contribution is to provide a unified quantum framework that achieves improvements over the best known classical bounds across a broad range of parameters $(n,m,t)$, while matching known bounds in extreme cases (e.g., $t = \Theta(n^3)$, where output size dominates).
>
> We thank the reviewer again for the constructive feedback.

---

> > ### Author Rebuttal · Reviewer_nM8w · 2026-04-02
> >
> > Thank you for your response. I believe most of my concerns have been addressed

---

### Official Review · Reviewer_WdpL · 2026-03-18

**Soundness:** 3
**Presentation:** 3
**Significance:** 2
**Originality:** 3
**Overall Recommendation:** 4
**Confidence:** 2

**Summary:**

This paper studies triangle cut sparsification, which aims to compress a graph while preserving triangle counts across all cuts, capturing higher-order structure beyond edges. The work analyzes how quantum algorithms can speed up this task, focusing on faster triangle listing as the main bottleneck. It introduces a new quantum triangle-listing algorithm with improved runtime over classical methods, and design a quantum algorithm that builds ε-triangle cut sparsifiers of near-optimal size more efficiently. They also prove a matching lower bound, showing this sparsifier size is essentially optimal.

**Compliance With Llm Reviewing Policy:**

Affirmed.

**Final Justification:**

My main question has been consistently addressed by the authors. I maintain my positive score.

**Key Questions For Authors:**

My question is about Theorem 1.2.At first sight, it is surprising to me that the size of the triangle cut sparsifier is O(n/\eps^2), independently of both m (edges) and/or (number of triangles).

Consider the case where (G) is a complete graph with unit weights:
  * For a balanced cut (|S| = n/2), there are c.n^3 triangles crossing the cut.
  * If the sparsifier (G') has only m = O(n/\eps^2) edges, then using the classical bound t \leq O(m^{3/2}), the number of triangles in (G') is at most O(n^{3/2}), which is to approximate the original number of crossing triangles.

How can the sparsifier preserve triangle cut values in such dense graphs? Is there a subtlety in the weighting or definition that resolves this apparent contradiction?

**Limitations:**

(yes)

**Strengths And Weaknesses:**

First, I would like to note that I am not familiar with quantum computing, so I cannot evaluate the quantum aspects, but rather give a fresh, outsider perspective focusing on clarity, positioning, and high-level understanding.

Strengths:
- This paper provides the first quantum bounds for triangle cut sparsification, which is a notable theoretical contribution.
- The literature review is strong and clearly motivates the importance of triangle-based structures in graphs.

Weakness:
- The paper relies heavily on quantum algorithms, which may limit accessibility and practical relevance for a broader ML audience. To me, is unclear whether this work fits ICML, and it may be more appropriate for a specialized algorithms venue, or a journal.
- The results are rather heavy to read, which make them difficult to follow for the non-expert, or even assess their novelty and significance. For instance, p.2 : "Our quantum algorithm accelerates the triangle-listing phase for a broad class of graphs" could be explicited : what kind of graphs? Is the complexity substantially better for dense, sparse, graphs ? All of them ?

---

> ### Author Rebuttal · Authors · 2026-03-31
>
> We sincerely thank the reviewer for the positive assessment and for the thoughtful questions on clarity and positioning. We address the main concerns below.
>
> **On ICML relevance and positioning.** The problem we study, triangle (motif) cut sparsification, is directly motivated by higher-order graph learning tasks such as triangle-based clustering and motif-aware community detection. These rely on efficient triangle listing and sparsification, and our work provides provable quantum speedups for these core primitives, improving the scalability of downstream ML pipelines. Methodologically, quantum algorithms naturally accelerate large combinatorial search problems (e.g., via Grover search and quantum walks). Our contribution is thus a systematic acceleration of key building blocks in graph-based learning, rather than a superficial use of quantum tools.
>
> Finally,  our work aligns with a growing line of research at the intersection of quantum algorithms and machine learning. Recent examples include quantum speedups for hypergraph sparsification at ICML 2025 (Liu et al., 2025 [1]) and related works (Liu et al., 2024 [2]; Peng & Xu, 2025 [3]), reflecting increasing interest in "quantum for AI". Our work extends this direction to triangle-based structures.
>
> **On regimes of improvement.** We thank the reviewer for pointing out that the regimes of improvement should be made more explicit. The best known classical bounds for triangle listing are $\widetilde{O}\left(n^\omega + n^{\frac{3(\omega-1)}{5-\omega}} t^{\frac{2(3-\omega)}{5-\omega}}\right)$ or  $\widetilde{O}\left(m^{\frac{2\omega}{\omega+1}} + m^{\frac{3(\omega-1)}{\omega+1}} t^{\frac{3-\omega}{\omega+1}}\right)$. Our quantum algorithm (Theorem 1.3) achieves polynomial improvements in several parameter regimes:
>
> - Sparse graph. For sparse graphs, our bound achieves matching or better dependence on $m$ and $t$ compared to the classical counterparts.
> - Moderate triangle regime. When $t \leq n^{15/14}$, our bound matches or improves the conjectured classical lower bound $\tilde{O}(n^2 + nt^{2/3})$ (assuming $\omega = 2$).
> - Dense triangle regime. For graphs with many triangles, the Grover-based term $\tilde{O}(n^{3/2}t^{1/2})$ yields improvements over classical bounds in certain regimes of $\omega$. For example, when $\omega > 7/3 \approx 2.333$, the time of our quantum algorithm is no larger than the classical bound for all $t \leq n^3$.
>
> **On Theorem 1.2.** We thank the reviewer for this insightful question. The key point is that we preserve *triangle cut values* in a *reweighted sparsifier*, not just the total number of triangles.
>
> The weight of each triangle is defined as the product of its edge weights, which generalizes the unweighted case (all weights $=1$) and can be interpreted as accounting for edge multiplicities. For a cut $(S,\bar S)$, the triangle cut value is the sum of weights of all triangles crossing the cut. In a complete graph with unit weights, a balanced cut has $\Theta(n^3)$ such triangles.
>
> This can be preserved by a sparse reweighted graph. Consider a (constant-degree) random $d$-regular graph where each edge has weight $\Theta(n/d)$. Although it contains far fewer triangles, each triangle carries weight about $(n/d)^3$. Intuitively, for a balanced cut, each vertex has about $d/2$ neighbors on the other side; choosing two such neighbors gives $\Theta(d^2)$ candidate pairs, and each pair forms a triangle with probability about $d/n$, yielding $\Theta(d^3/n)$ crossing triangles per vertex and thus $\Theta(d^3)$ in total for the cut. After reweighting, the triangle cut value becomes $\Theta(d^3\cdot(n/d)^3)=\Theta(n^3)$.
>
> Thus, the sparsifier preserves the *weighted triangle cut values* even though the total number of triangles is much smaller. The apparent contradiction comes from comparing counts with cut values; indeed the latter are preserved.
>
> We thank the reviewer again for the helpful feedback.
>
> [1] Liu, C., Gao, M., Ji, Z., & Ying, M. *Quantum speedup for hypergraph sparsification.* ICML 2025.
> [2] Liu, C., Guan, C., He, J., & Lui, J. C. *Quantum algorithms for non-smooth non-convex optimization.* NeurIPS 2024.
> [3] Peng, P., & Xu, H. *Differentially private synthetic graphs preserving triangle-motif cuts.* COLT 2025.

---

> > ### Author Rebuttal · Reviewer_WdpL · 2026-04-07
> >
> > My main question has been consistently addressed by the authors. I maintain my positive score.

---

### Decision · Program_Chairs · 2026-04-30

**Decision:**

Accept (regular)

**Comment:**

The paper proposes quantum speed-ups for triangle cut sparsification via a careful examination and combination of several quantum components. There is a census about the contribution of this submission. The authors are encouraged to revise based on reviewers’ feedback.